# Compile to Compress: Boosting Formal Theorem Provers by Compiler Outputs

Guchan Li[1]   Rui Tian[1]   Hongning Wang[1]

## Abstract

Large language models (LLMs) have demonstrated significant potential in formal theorem proving, yet state-of-the-art performance often necessitates prohibitive test-time compute via massive roll-outs or extended context windows. In this work, we address this scalability bottleneck by exploiting an informative structure in formal verification: the observation that compilers map a vast space of diverse proof attempts to a compact set of structured failure modes. We introduce a learning-to-refine framework that leverages this compression to perform efficient learning and proof exploration. We perform tree search that corrects errors locally conditioned on explicit verifier feedback, thereby circumventing the costs associated with accumulating a long history of proof attempts. Extensive evaluations show that our method consistently amplifies the reasoning capabilities of base provers across varying scales. Notably, our approach achieves state-of-the-art performance on PutnamBench among publicly reported ∼8B and ∼32B parameter models under comparable test-time budgets, offering a scalable paradigm for next-generation verifier-guided reasoning.

## 1. Introduction

At the intersection of mathematical rigor and computational precision, formal theorem proving has become a cornerstone of humanity's pursuit of reliable and scalable reasoning. Within the unified framework of the Curry–Howard correspondence (Howard, 1980), mathematical proofs can be casted as programs, and proof verification reduces to type checking. This paradigm enables mathematical correctness

[1]Department of Computer Science and Technology, Tsinghua University, Beijing, China. Correspondence to: Guchan Li <li-gc22@mails.tsinghua.edu.cn>, Rui Tian <tianr22@mails.tsinghua.edu.cn>, Hongning Wang <hw-ai@tsinghua.edu.cn>.

*Proceedings of the 43rd International Conference on Machine Learning*, Seoul, South Korea. PMLR 306, 2026. Copyright 2026 by the author(s).

to be enforced with machine-level certainty. In recent years, modern proof assistants—most notably the Lean programming language (Moura & Ullrich, 2021)—have successfully bridged human mathematical intuition with mechanically verifiable programs, unlocking new possibilities for large-scale formalization.

The rapid integration of large language models (LLMs) into theorem proving has further accelerated progress in this domain. Contemporary Lean provers span a broad methodological spectrum, including: (1) supervised fine-tuning (SFT) on synthetic proof data (Xin et al., 2024a), (2) reinforcement learning (RL) to enhance chain-of-thought (CoT) reasoning in formal proofs (Wang et al., 2025; Ren et al., 2025), (3) self-correction driven by compiler feedback (Lin et al., 2025b), and (4) large-scale pipelines (Chen et al., 2025b) or agentic systems (Breen et al., 2025).

While compiler feedback consistently drives performance, existing approaches often oversimplify these signals into binary success indicators (Lin et al., 2025a) or plain-text (Yang et al., 2023) error messages. This neglects the rich structural information embedded in compiler outputs, forcing a reliance on cumulative, multi-round self-correction. Consequently, the model remains tethered to long interaction histories, causing context windows to expand rapidly and severely constraining the depth of exploration within fixed computational budgets.

By analyzing the failure modes of existing provers (see Section 3 and Appendix A), we identify that Lean compiler acts as an effective dimension compressor. As illustrated in Figure 1 (a), while the combinatorics of Lean programs render the input space effectively unbounded, the compiler projects these inputs into a compact, structured output space. This projection reveals an informative latent structure: many syntactically distinct incorrect proofs induce invariant error signatures. This suggests that compiler feedback can filter out syntactic noise to expose the underlying semantic defects. Leveraging this compressed signal allows us to shift the learning objective from blind generation to targeted correction, exploiting the structured feedback to navigate the proof space more efficiently.

Building on this insight, we introduce a novel learning-to-refine framework that internalizes error-correction trajectories into an LLM via supervised fine-tuning and expert

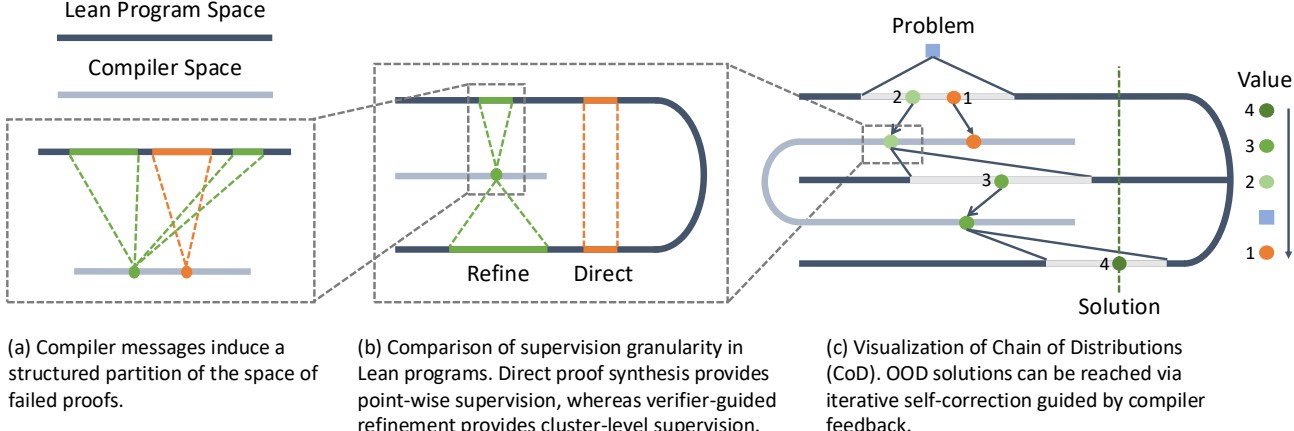

(a) Compiler messages induce a structured partition of the space of failed proofs.

(b) Comparison of supervision granularity in Lean programs. Direct proof synthesis provides point-wise supervision, whereas verifier-guided refinement provides cluster-level supervision.

(c) Visualization of Chain of Distributions (CoD). OOD solutions can be reached via iterative self-correction guided by compiler feedback.

*Figure 1.* Overview of the proposed learning-to-refine framework. Compiler messages map the space of diverse proof attempts to a compact set of structured failure modes, which enable efficient learning and search for the correct proofs. We trained a neural value model to efficiently guide test-time proof generation and refinement process.

iteration. This effectively distills complex formal reasoning capability into a generalizable form of policy that can both generate and refine proofs. Recurrently applying this refinement skill during multi-round self-correction, a chain of distributions (CoD) emerges in program space. Combined with a learned search strategy, this adaptive exploration mechanism achieves strong performance across multiple benchmarks. Moreover, our framework is generally model-agnostic and scales consistently with model sizes and test-time budget, offering a scalable paradigm for next-generation verifier-guided reasoning.

## 2. Related Work

Lean 4 programming language (Moura & Ullrich, 2021) represents a new practical paradigm for mathematics. It transfers step-by-step human-judged math deductions into a structured, human-machine interaction process. Its ultimate value include enhanced rigor, high reusability of mathematical knowledge, and the potential to open new pathways for mathematical discovery.

Along with the development of deep learning, initial attempts on automated theorem proving (ATP) focus on efficient tree search algorithms (Kaliszyk et al., 2018; Lample et al., 2022; Polu et al., 2022) and proof-step generation (Han et al., 2022; Yang et al., 2023). Due to the lack of effective semantic representations, these methods suffer from huge search space and intensive computation cost.

Recent efforts utilize LLMs to facilitate the process (First et al., 2023). Early work mostly focuses on synthesizing large-scale Lean 4 proof data to fine-tune LLMs (Xin et al., 2024a; Wang et al., 2024; Lin et al., 2025a; Dong & Ma, 2025). Later, following the success of reinforcement learning (RL) in natural language reasoning (Shao et al., 2024;

Guo et al., 2025), similar approaches have been applied to formal reasoning (Xin et al., 2024b; Ren et al., 2025; Wang et al., 2025; Lin et al., 2025b), where Lean compiler serves as the reward signal and environment feedback. Prior to our work, compiler messages have been incorporated into LLM input (First et al., 2023; Lin et al., 2025b) for improved ATP. But existing approaches are constrained by growing context complexity and roll-out cost, thus failing to fully leverage the potential of compiler feedback.

Our work focuses on exploiting compiler feedback for complete proof generation. By viewing the Lean compiler as a dimension compressor, we demonstrate how this perspective significantly enhances the efficiency of self-correction in formal theorem proving.

In parallel, researchers also looked into building large theorem proving systems (Chen et al., 2025b;a; Logical Intelligence, 2025) or agent systems (Breen et al., 2025; Varambally et al., 2025; Requena et al., 2026). Currently, these systems achieve state-of-the-art performance on various benchmarks, but they depend on massive models and intense inference cost, hence not directly comparable to ours.

## 3. Lean Compiler as a Dimension Compressor

Table 1 reports the most frequent compilation error messages produced by the proofs generated by Kimina-Prover-Preview-7B (Wang et al., 2025) on MiniF2F-test (Zheng et al., 2022) (244 problems) and PutnamBench (Tsoukalas et al., 2024) (660 problems). To make the structure of messages more visible, variable names, tactic states, and other context-dependent information are masked or removed. We can clearly observe that a small number of distinct compiler messages account for a majority of failures across both benchmarks.

*Table 1.* Top 50% error messages associated with proofs produced by Kimina-Prover-Distill-8B on standard benchmarks. Identifiers are masked by 'id'. Appendix A reports the full table.

| MINIF2F-TEST | RATIO |
|---|---|
| LINARITH FAILED TO FIND A CONTRADICTION | 22.60% |
| UNKNOWN IDENTIFIER 'ID' | 16.31% |
| UNSOLVED GOALS | 13.58% |
| PUTNAM BENCH | |
| TACTIC 'ID' FAILED, NESTED ERROR | 16.78% |
| OMEGA COULD NOT PROVE THE GOAL | 13.50% |
| UNSOLVED GOALS | 11.31% |
| LINARITH FAILED TO FIND A CONTRADICTION | 8.10% |
| UNKNOWN IDENTIFIER 'ID' | 5.59% |

This many-to-one projection induced by compilation process could be viewed as a compression operator: diverse incorrect proofs are mapped to a finite set of diagnostically meaningful failure modes. By inducing a low-entropy partition over failure modes, Lean compiler successfully makes learning a conditional refinement policy statistically more feasible.

### 3.1. Notations

Let $P$ denote the space of problems, $C$ denote the space of Lean 4 programs, and $M$ denote the space of compiler messages. Let

$$r^* : P \times C \to C$$

denote the ideal refinement function that fixes an incorrect Lean program to a correct one. In verifier-guided self-correction, verifier is modeled as a function,

$$\Phi : C \to M.$$

Empirically, $\Phi$ is a many-to-one mapping: distinct Lean programs may induce identical compiler feedback. This induces a natural partition of program space, where programs that produce the same compiler error message are grouped. For program $c$, we define

$$\Psi(c) := \Phi^{-1}(\Phi(c)) = \{c' \in C \mid \Phi(c') = \Phi(c)\}.$$

as the set of programs associated with the same compiler feedback. With Lean compiler, each program $c$ induces a supervised signal $S(c)$ in the following form,

$$S(c) = \begin{cases} p \mapsto c & \text{if } c \text{ is correct} \\ (p, c) \mapsto r^*(p, c) & \text{if } c \text{ is incorrect} \end{cases}$$

Correct programs therefore induce synthesis supervision (i.e., a verified proof of problem $p$), whereas incorrect programs induce refinement supervision (i.e., reparation of problem $p$'s incorrect proof $c$). Both are unified under the same supervision operator $S$, but differ fundamentally in their structure and information conveyed.

### 3.2. Failure-Mode Driven Refinement

Our key insight behind verifier-guided self-correction is that programs failing for similar reasons often admit similar refinement strategies. Ideally, the refinement target $r^*(p, c)$ should depend on the underlying failure modes, rather than superficial syntactic details of the program. We refer to this inductive bias as failure-mode driven refinement.

In practice, we observe the projections of failure modes through Lean compiler output $\Phi(c)$. Although compiler messages are lossy and do not capture full semantic context of the mathematic derivations, they frequently encode high-level structural information about why a proof attempt failed (e.g., unsolved goals, type mismatches, or tactic failures). As a result, syntactically distinct programs that induce the same compiler message often benefit from similar corrective actions.

This perspective highlights a fundamental difference between direct solution synthesis and compiler-guided refinement. As illustrated in Figure 1 (b), direct supervision provides point-wise targets in the program space, tightly coupling learning to individual problem instances. In contrast, verifier-guided refinement conditions on compiler feedback, which induces a coarse partition of program space and encourages the learning of refinement strategies that generalize across programs exhibiting similar failure patterns.

### 3.3. Chain of Distributions

Under the view of the Lean compiler as a dimension compression operator, correcting a failed Lean program $c$ implicitly leverages refinement supervision $S(c')$ collected from training programs $c'$ that exhibit similar compiler feedback, i.e., $c' \in \Psi(c)$. During multi-round self-correction, the prover repeatedly invokes refinement knowledge learned from programs that may originally target at distinct problems or follow different distributions in the program space. As a result, the refinement process induces an adaptive trajectory over the distributions of programs rather than sampling from a single fixed distribution.

Formally, the self-correction process defines a chain of conditional distributions (CoD) over the program space:

$$\{D_{\text{refine}}(\cdot \mid c_i, \Phi(c_i), p)\}_{i=1}^n$$

where each subsequent program $c_{i+1}$ is sampled conditioned on the previous refinement state $s_i = (c_i, \Phi(c_i), p)$ consisting previous failed attempt $c_i$, the error message $\Phi(c_i)$ and the problem $p$. This chain evolves dynamically as refinement progresses and can be viewed as a bridge between the initial generation distribution and regions in the program space that contain correct solutions.

In contrast, direct proof generation samples programs from a single, fixed distribution $D_{\text{direct}}(\cdot \mid p)$ independent of

intermediate feedback. If correct solutions lie far from the high-probability mass of $D_{\text{direct}}(\cdot \mid p)$, they are effectively out-of-distribution (OOD), and discovering them requires a prohibitively large number of samples.

As illustrated in Figure 1 (c), direct proof generation remains confined to the support of a static distribution, whereas the refinement process iteratively reshapes its sampling distributions through feedback-conditioned updates. This adaptive process enables the model to progressively move toward correct proofs that may initially lie outside the support of $D_{\text{direct}}(\cdot \mid p)$, providing a principled mechanism for escaping local modes in the program space.

# 4. Boosting Lean Provers with a Markovian Refining Process

In this section, we boost a given Lean prover into an adaptive refinement-augmented prover. Under the chain-of-distributions (CoD) framework, each refinement step only depends on the current state, because historical refinement attempts are implicitly encoded in Lean semantics. This Markovian treatment not only removes burden on LLM context lengths, but also leads to a simplified training procedure.

Generally, existing chain-of-thought (CoT) provers generate responses in a "thought + program" format (Wang et al., 2025; Lin et al., 2025b). Accordingly, each refinement training instance is constructed as follows: the input is the refinement state $s_i$, the output contains analysis of failure messages and the corrected Lean program. We combine Lean program $c$ and compiler messages $\Phi(c)$ using special tokens based on line number in compiler messages. Details, ablation studies and examples are reported in Appendix B.1 and D.1.

## 4.1. Cold-Start Data Synthesis

Initially, the Lean prover does not have self-correction ability. We design the data synthesis pipeline illustrated in Figure 2 to collect refinement training data. For each problem $p$, we sample multiple responses from the prover, collecting both an incorrect and a correct Lean solution $(c_{incorrect}, c_{correct})$. We construct the refinement function using a synthesized thought $t$:

$$r^*(p, c_{incorrect}) \leftarrow (t, c_{correct})$$

To populate the "thought" component $t$ in the response, we use Claude-3.7-Sonnet to analyze compiler errors and generate a detailed refinement plan that leads from the failed proof to the correct solution. Our training dataset is sourced from Goedel-LM/Goedel-Pset-v1 (Lin et al., 2025a), and details of our dataset construction and employed prompts are reported in Appendix B.2.

Our proposed data synthesis pipeline is hallucination-free

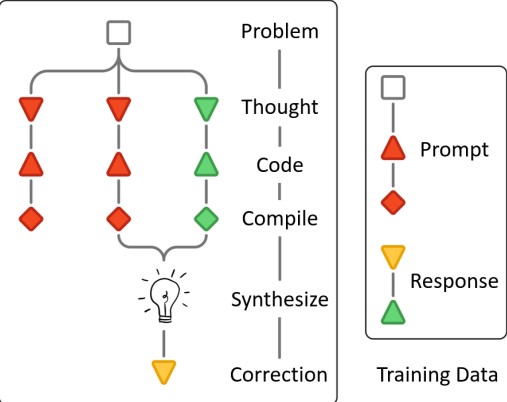

*Figure 2.* Refinement training data synthesis pipeline. Experiments were conducted using Kimina Lean server (Santos et al., 2025), with Lean v4.15.0 and a pinned snapshot of Mathlib.

and cost-effective. All corrected solutions are validated by the Lean compiler, and the refinement data is derived directly from the failures of the prover itself. To preserve the prover's original problem-solving ability, we also include directly solved solutions as direct training data. Since the amount of direct problem solving data exceeds that of refinement data, we randomly subsample the direct training data to match the size of refinement data.

Supervised fine-tuning the prover on this collected refinement data enables it to reliably repair its own failed attempts, leading to improved robustness without introducing distributional mismatch. To separate the contribution of refinement learning and Claude-generated traces, discussions and ablations are included in Appendix C.3.

## 4.2. Expert Iteration

Once the prover starts to refine its proofs, the distribution of failed attempts drifts (see Figure 1 (c)). Hence, to fully exploit the potential of CoD, it is important to continuously improve the prover's refinement ability along the way. In this work, we choose expert iteration for the purpose.

We implement expert iteration by executing the prover on a corpus of training problems and iteratively refining its last failed proof attempt, until a correct solution is obtained or the budget is exhausted. Successful trajectories are then converted into refinement supervision, while problems solved in a single step are retained as direct generation instances. As in the SFT training, we maintain a balanced mixture of direct and refinement data. The resulting dataset therefore realizes on-policy training of the prover. This is crucial for a self-refining prover: such training progressively adapts the policy's refinement ability to its own evolving error distribution, enabling the prover to learn repair strategies for failure modes that only emerge after earlier refinements.

## 4.3. Test-Time Search Strategy

At test time, the prover can choose to either generate a new proof attempt from scratch, or select a previously failed program from the history for refinement. In any case, the prover generates a new Lean proof. We model this iterative self-correction process as a search problem. From a search perspective, direct proof attempts correspond to breadth-first search (BFS), as they explore multiple independent proofs for the target problem as the root node. Meanwhile, refining a previous failed attempt (as an internal tree node) correspond to depth-first search (DFS). How to adapt between these two strategies becomes the key to the success of our learning-to-refine framework.

### 4.3.1. RANDOM TREE SEARCH

As a starting point, we consider random tree search, where at each step a node is sampled uniformly from the current tree and expanded. Selecting the root generates a new proof attempt, while selecting an internal node triggers refinement. This process constructs a random recursive tree, whose expected width and depth are both $O(\log n)$ with $n$ nodes (Pittel, 1994). Intuitively, this yields a balanced mixture of directly solving the problem and refining previous attempts as the computational budget increases.

### 4.3.2. VALUE-GUIDED TREE SEARCH

Random tree search implicitly assumes that all nodes are equally promising, whereas in practice different proof attempts exhibit highly heterogeneous potential for leading to a verified proof. To better allocate computational budget, we introduce a neural value function that estimates the relative potential of different choices in a search tree. The backbone model architecture is based on our refinement-augmented prover after the expert iteration step, and details are reported in Appendix B.5. The value function takes the same prompt $s_i$ as input, and outputs a scalar value suggesting the potential of target node.

We follow standard preference learning framework for value function estimation (Christiano et al., 2023), in which supervision is constructed through pairwise comparisons among refinement states $\{s_i = (c_i, \Phi(c_i), p)\}$, where $c$ is the current proof and $p$ is the problem. Let $\tau = (s_0, s_1, \ldots, s_T)$ denote a successful refinement trajectory for problem $p$, where $s_0 = p$ is the root node and $s_T$ yields a verified proof.

As shown in Figure 1 (c), preferences are constructed in two ways, i.e., 1) by ordering states along successful refinement paths, and 2) by comparing the root state against states on failed trajectories:

1. For any two states $s_i$ and $s_j$ on the same successful

*Table 2.* Summarization of experiments and training data quantity. *<Prover>Expert* denotes our main experiment. *<Prover>Direct* corresponds to direct synthesis training as comparison.

| MODEL | TRAIN DATA | STRATEGY |
|---|---|---|
| KIMINA EXPERT | 19.4K | BFS, DFS, TREE |
| GOEDEL EXPERT | 22.4K | BFS, DFS, TREE |
| KIMINA DIRECT | 37K | BFS |
| GOEDEL DIRECT | 43K | BFS |

    trajectory with $i > j$, we define $s_i \succ s_j$, reflecting that states closer to a verified proof are empirically more promising for further refinement. This captures relative progress along a refinement path.

2. For every non-root state $s_k \neq s_0$ outside the successful trajectories, we define $s_0 \succ s_k$, encouraging the model to generate new proofs from scratch than to refine a failed attempt whose subsequent refinements never lead to a success.

Together, these pairwise comparisons enable the value function to capture both local refinement progress and global refinement potential.

Value function training is performed iteratively. Starting from random tree search, we collect preference pairs to train the initial value function $V_\theta^{(0)}$. In each subsequent round $k$, the latest value function $V_\theta^{(k)}$ guides tree search to generate new trajectories, which are then incorporated into the training of $V_\theta^{(k+1)}$. Specifically, value-guided tree search expands tree nodes according to a softmax distribution over the predicted values:

$$\pi_\theta(c \mid p) \propto \exp(V_\theta^{(k)}(c, \Phi(c), p))$$

This stochastic search policy ensures non-zero probability for all nodes, guaranteeing asymptotic coverage of the search tree, while exponentially biasing exploration toward regions with higher predicted refinement potential. As a result, the iterative training process progressively exposes the value function to more informative refinement paths that are unlikely to be discovered under random tree search.

## 5. Experiment

### 5.1. Provers and Experiment Settings

We applied our learning-to-refine framework to Kimina-Prover-Distill-8B (Wang et al., 2025) and Goedel-Prover-V2-32B (Lin et al., 2025b), which roughly represent the performance of the lower and upper bounds of medium-scale Lean provers. Although Goedel-V2 supports refinement internally, we only employ its direct solution generation ability, because it does not satisfy the Markovian refinement

*Table 3.* Accuracy on benchmarks with sampling budget of 64. For PutnamBench, we report the number of passed problems. *<Prover>Direct* denotes base provers fine-tuned with directly solved examples. *<Prover>Random* and *<Prover>Value* correspond to different test-time search strategies. Columns *Putnam 128 / 256* report scaling effects on PutnamBench.

| METHOD | MINIF2F-TEST | PROOFNET | MOBENCH | PUTNAM | PUTNAM 128 | PUTNAM 256 |
|---|---|---|---|---|---|---|
| KIMINA | 77.46 | 14.56 | 7.78 | 10 | 10 | 10 |
| KIMINA DIRECT | 75.41 | 14.82 | 7.78 | 12 | 14 | 16 |
| KIMINA RANDOM | 81.15 | **15.63** | **8.61** | 17 | 20 | 21 |
| KIMINA VALUE | **81.97** | 15.36 | **8.61** | **20** | **23** | **25** |
| GOEDEL-V2 | 84.43 | 15.63 | 14.72 | 32 | 38 | 41 |
| GOEDEL DIRECT | 84.84 | 19.14 | 12.50 | 28 | 37 | 44 |
| GOEDEL RANDOM | 84.43 | 23.72 | 30.28 | **63** | 80 | 104 |
| GOEDEL VALUE | **86.89** | **24.26** | **34.44** | **63** | **82** | **110** |

property. Results of directly applying random tree search to Goedel-V2 are reported in Table 16.

As summarized in Table 2, our experiments disentangle two orthogonal dimensions of evaluation: (1) the form of supervision used during training (direct synthesis or refinement), and (2) the strategy used to allocate test-time sampling budget (BFS, DFS, random or value-guided tree search).

For Kimina-8B, we collected 3.1k and 6.6k refinement instances across two training stages. For Goedel-32B, the corresponding sizes were 4.5k and 6.7k. As explained previously, we added an equal amount of directly solved problems, resulting in 19.4k and 22.4k training data instances for Kimina and Goedel-V2 respectively. The resulting models after expert iteration are denoted as Kimina-Expert and Goedel-Expert. For comparison purposes, we utilized all the directly solved problems produced by base provers to fine-tune themselves (denoted as Kimina-Direct and Goedel-Direct in our experiments).

### 5.2. Evaluation Results on Benchmarks

We evaluated different provers on four benchmarks:

- MiniF2F (Zheng et al., 2022) consists of 488 problems in Lean 4 (244 for validation and 244 for test). This dataset covers high-school level math problems, including AIME, AMC and IMO competitions. We use the test split for evaluation.

- ProofNet (Azerbayev et al., 2023) gathers 371 problems from undergraduate-level mathematics textbooks, covering real and complex analysis, linear algebra, abstract algebra and topology. We use the Lean 4 version provided by DeepSeek-Prover-V1.5 (Xin et al., 2024b).

- MathOlympiadBench is introduced in the work of Goedel-Prover-V2 (Lin et al., 2025b). It comprises 360 olympiad-level mathematical problems.

- PutnamBench (Tsoukalas et al., 2024) contains 660 problems drawn from the Willian Lowell Putnam Mathematical Competition between years 1962-2024.

*Table 4.* The PutnamBench leaderboard. We report top methods' model parameter, test-time budget and number of solved problems. Our methods are highlighted in bold. Methods that utilize compiler messages are marked with asterisk.

| METHOD | SIZE | BUDGET | NUM-SOLVED |
|---|---|---|---|
| ALEPH | ? | $1400 | 668 |
| ALEPH | ? | $400 | 637 |
| SEED-V1.5* | ? | 10H 20DAYS | 581 |
| ALEPH | ? | $100 | 500 |
| HILBERT* | ? | 1840 | 462 |
| AX-BASE* | ? | AVG. $12.6 | 365 |
| SEED* | ? | MEDIUM | 329 |
| **OURS*** | **32B** | **256** | **110** |
| AX-PROVER* | $\geq$ 72B | 100 TOOLS | 91 |
| GOEDEL-V2* | 32B | 184 | 86 |
| DSP-V2 | 671B | 1024 | 47 |
| GPT-5* | ? | 10 TOOLS | 28 |
| DSP+* | $\geq$ 32B | 128 | 23 |
| **OURS*** | **8B** | **256** | **25** |
| BOURBAKI | 7B | 512 | 14 |
| KIMINA | 8B | 192 | 10 |
| STP | 7B | 3200 | 8 |
| GOEDEL-V1 | 7B | 512 | 7 |

While certain other works have included problems in year 2025, we excluded them to align with the settings in the original papers of our baseline methods for consistency.

The comparison results are reported in Table 3. Compiler-guided self-correction show substantial advantage across provers and benchmarks. Kimina-Expert and Goedel-Expert outperform their base counterparts on all benchmarks, reaching over 50% relative improvements on ProofNet, MathOlympiadBench and PutnamBench in Goedel-V2. Value-guided tree search further improved accuracy, surpassing the random search baselines on most test cases.

As shown in Table 4, we successfully boosted base provers to state-of-the-art performance among provers of the same model size on PutnamBench. In the ~32B regime, our Goedel-Expert model solved 110 problems, outperforming all other solutions of comparable scale, including Ax-Prover and original Goedel-V2 (in its self-correction mode). In

*Table 5.* Difficulty breakdown of the MiniF2F-test benchmark with sampling budget 64. *Kimina Ratio* and *Goedel Ratio* represent the ratio of corresponding splits under Kimina-Prover-Distill-8B and Goedel-Prover-V2-32B.

|  | EASY | MEDIUM | HARD |
|---|---|---|---|
| KIMINA RATIO | 67.62 | 9.83 | 22.54 |
| BFS | 99.40 | 91.67 | 10.91 |
| DFS | 100 | 91.67 | **20.00** |
| RANDOM | 100 | **95.83** | 18.18 |
| VALUE | 100 | 91.67 | **20.00** |
| GOEDEL RATIO | 73.77 | 10.66 | 15.57 |
| BFS | 100 | 76.92 | 10.52 |
| DFS | 97.78 | 61.54 | 13.16 |
| RANDOM | 99.44 | 76.92 | 18.42 |
| VALUE | 100 | **84.61** | **26.32** |

the $\sim$8B regime, our Kimina-Expert model solved 25 problems, again ranking the first among provers of similar size. We also report performance of close-sourced large theorem proving systems, whose computational budgets were not measured under the same standard. Together with the results in Table 3, our solution exhibits promising scaling effect with respect to both model size and test-time computation.

Despite fine-tuned with more directly solved problems, Kimina-Direct and Goedel-Direct exhibit only marginal or inconsistent improvements across benchmarks. This contrast indicates that the gains of refinement cannot be attributed to additional training data or standard supervised fine-tuning alone. Instead, refinement supervision signals propagate across equivalence classes of failed proof attempts defined by the failure modes, enabling transfer across problems that share similar error structures, which direct supervision cannot capture.

## 5.3. Comparison of Test-Time Strategies

The limited gains on MiniF2F-test should not be viewed as a failure of refinement, but rather as a regime where direct generation is already near-saturated. Let $P_{\text{direct}}(p)$ denote the probability of solving problem $p$ via direct generation, and let $P_{\text{refine}}(c, p)$ denote the probability that a failed program $c$ can be successfully repaired. Given sampling budget $n$, a direct prover generates $n$ independent candidates, and the probability of solving $p$ is

$$P_1(p) = 1 - (1 - P_{\text{direct}}(p))^n$$

Under the random tree search strategy, the expected degree of root node is the harmonic number $H_n$. For analytical simplicity, we assume that $P_{\text{refine}}(c, p)$ is uniformly bounded by $P_{\text{refine}}(p)$, independent of the specific failed proof $c$. The marginal probability of solving $p$ using refinement is then

$$P_2(p) = 1 - (1 - P_{\text{direct}}(p))^{H_n} \times (1 - P_{\text{refine}}(p))^{n - H_n}$$

*Table 6.* Distributional hypothesis testing on the MiniF2F-test-hard split. Each cell reports the ratio of problems satisfying the p-value threshold. *#problem* reports the number of problems in each split.

| P-VALUE | $\leq 0.05$ | $\leq 0.005$ | $\leq 0.001$ | #PROBLEM |
|---|---|---|---|---|
| KIMINA | 53 | 45 | 39 | 69 |
| GOEDEL | 43 | 41 | 38 | 52 |

For refinement to outperform direct proof generation, we require $P_2(p) > P_1(p)$, which can be simplified to

$$P_{\text{refine}}(p) > P_{\text{direct}}(p)$$

This analysis provides one central message: refinement is not omnipotent everywhere. For many problems in MiniF2F-test, $P_{\text{direct}}(p)$ is sufficiently large to void the above inequality. To demonstrate this, we further stratified MiniF2F-test by problem difficulty. A problem is labeled as **easy** if the base prover (Kimina or Goedel-V2) solves it in $\leq 4$ sampling passes, **medium** if it is solved with $5 \sim 64$ passes, and **hard** otherwise. We compare BFS (direct sampling), DFS (iterative refinement), random tree search and value-guided tree search strategies on these three types of problems accordingly.

Results are summarized in Table 5. BFS performs well on **medium** problems, where direct solutions are often reachable within a few sampling rounds, while DFS is more effective on **hard** problems, where iterative error correction is necessary. Random tree search provides a balanced trade-off, consistently outperforming both BFS and DFS by combining the two strategies. Value-guided tree search further improves efficiency by prioritizing branches with higher estimated potential, yielding stronger performance overall, especially on hard problems.

## 5.4. Analysis of Chain of Distributions

Refinement process induces a sequence of conditional distributions that evolve based on intermediate compiler feedback. In this section, we empirically test whether unconditioned generation and compiler-conditioned refinement indeed correspond to distinct distributions over the Lean program space. Concretely, programs produced by BFS are samples from $D_{\text{direct}}(\cdot \mid p)$, and those generated from the descendants of the root node by random tree search are samples from the aggregated distribution $\{D_{\text{refine}}(\cdot \mid c_i, \Phi(c_i), p)\}$. For simplicity, we analyze the overall refinement distribution rather than step-level transitions.

To compare these distributions, we apply an energy-based two-sample hypothesis test, where the null hypothesis is that the two sets of samples are drawn from the same distribution. We use string-level edit distance as a computable proxy for syntactic proximity, leveraging the fact that Lean's syntax is tightly coupled to proof structure and localized semantic changes. Appendix C.1 reports detailed procedures.

*Table 7.* Top-10 errors produced by Goedel-Expert under random search strategy. Frequencies in training set are reported in Appendix B.4. The results of Kimina-Expert are reported in Appendix C.2.

| ERROR TYPE | OCCUR FREQ (%) | #PROBLEM | FIX PROB (%) | TRAIN FREQ (%) |
|---|---|---|---|---|
| UNSOLVED GOALS | 23.83 | 31 | 0.34 | 16.41 |
| LINARITH FAILED TO FIND A CONTRADICTION | 19.79 | 28 | **0.07** | **25.41** |
| OMEGA COULD NOT PROVE THE GOAL: | 10.92 | 20 | 0.12 | 7.78 |
| TACTIC 'ID' FAILED, NESTED ERROR | 3.14 | 12 | **0.00** | **3.08** |
| MAXIMUM RECURSION DEPTH HAS BEEN REACHED | 2.86 | 15 | 0.00 | 0.00 |
| UNKNOWN IDENTIFIER 'ID' | 1.93 | 13 | **1.38** | **8.51** |
| TYPE MISMATCH | 1.65 | 20 | 0.81 | 4.23 |
| FAILED TO SYNTHESIZE | 1.32 | 11 | **4.04** | **2.30** |
| UNKNOWN CONSTANT 'ID' | 1.24 | 11 | 1.08 | 1.49 |
| TACTIC 'ID' FAILED, FAILED TO UNIFY | 1.13 | 11 | 0.00 | 0.00 |

Table 6 shows the fraction of problems on MiniF2F-test-hard split for which the null hypothesis is rejected. For both Kimina-Expert and Goedel-Expert, the null hypothesis is rejected for a majority of problems, indicating that refinement induces a statistically distinct sampling distribution from direct solution generation. This analysis provides quantitative evidence that refinement explores regions of program space that are unlikely to be reached by direct sampling alone, consistent with our CoD hypothesis.

### 5.5. Error-Type Analysis

We further analyzed the prover's refinement behavior on the MiniF2F-test-hard split. Table 7 reports, for the most frequent error types produced by Goedel-Expert, their occurrence frequency, the number of distinct problems in which they appear, the empirical probability of being fixed, and their frequency of appearance in the training data.

The probabilities are computed at error-type level. When multiple errors occur in one instance, each error is treated independently, and the correction probability is defined as the fraction of occurrences resolved. Each error type's probability of correction is aggregated across problems, and does not measure transition tree nodes. Similar statistics on Kimina-Expert are reported in Appendix C.2.

Several patterns emerge. First, difficulty for correction varies substantially across error types. Errors such as `failed to synthesize` admit non-negligible correction rates, while others, including `linarith failed`, remain difficult to repair despite occurring frequently in the training set. This indicates that semantic complexity still remains a limiting factor for refinement.

Second, all error types with nonzero correction probability are present in the training data and appear across many distinct problems, suggesting that refinement generalizes across recurring failure modes rather than simply memorizing problem-specific solutions. However, training exposure alone is insufficient for several error types (e.g. `nested error`): they occur frequently in training, but still exhibit

near-zero correction probability at test time. These errors correspond to global or poorly localized failures, for which reusable local repair strategies may not exist.

Overall, this analysis supports the view that compiler feedback induces a coarse but meaningful partition of failure modes. Refinement learning is most effective when applied to recurring, structurally regular errors that admit localized corrections.

### 5.6. Computation Cost Analysis

The computational cost of our framework consists of three components:

- **Lean compilation cost.** Each sampled proof requires exactly one compilation call, making the total number of compilations comparable to direct proof generation approaches. In practice, compilation overhead is negligible relative to model inference due to the efficiency of the Kimina Lean server.

- **Value model inference cost.** The value model introduces additional memory usage, but its inference latency is negligible compared to autoregressive decoding, accounting for less than 0.1% of total runtime.

- **Token generation cost.** This constitutes the dominant computational expense. We measure token usage for Goedel-Prover-V2 under direct sampling and Goedel-Expert under random tree search with refinement. Results are summarized in Table 8.

Although refinement introduces longer input sequences due to injected compiler feedback, it substantially reduces both average and total output length across benchmarks. Since inference cost is dominated by autoregressive decoding, this reduction leads to significantly lower overall generation cost.

This behavior is consistent with our context-light formulation: failed Lean programs together with compiler feedback

*Table 8.* Token usage for Goedel-V2 and Goedel-Expert across benchmarks.

| DATASET | MINIF2F-TEST | PROOFNET | MOBENCH | PUTNAMBENCH |
|---|---|---|---|---|
| BUDGET | 64 | 64 | 64 | 256 |
| V2 AVG. INPUT | 271.30 | 157.61 | 211.99 | 336.49 |
| V2 AVG. OUTPUT | 6267.47 | 7415.36 | 12200.99 | 10145.58 |
| EXPERT AVG. INPUT | 2799.80 | 2208.98 | 3230.62 | 3560.39 |
| EXPERT AVG. OUTPUT | 3538.73 | 3016.76 | 4560.31 | 3555.41 |
| V2 TOTAL INPUT | 843 K | 3,253 K | 4,260 K | 53,869 K |
| V2 TOTAL OUTPUT | 8,197 K | 122,909 K | 142,788 K | 924,911 K |
| EXPERT TOTAL INPUT | 8,349 K | 53,980 K | 41,188 K | 509,843 K |
| EXPERT TOTAL OUTPUT | 9,834 K | 53,755 K | 73,098 K | 476,133 K |

implicitly preserve previous reasoning progress, thereby reducing the need for long exploratory reasoning traces during subsequent refinement steps.

Importantly, the Markovian refinement formulation removes dependence on the full refinement history. As a result, input length remains approximately constant even as the number of refinement rounds increases. In contrast, non-Markovian refinement strategies accumulate historical context linearly with the number of iterations, leading to substantially higher inference cost.

## 6. Conclusion and Future Work

In this work, we presented a learning-to-refine framework for formal theorem proving that exploits a key structural property of verifier-guided reasoning: diverse proof attempts are mapped by the compiler to a compact set of structured failure modes. Treating compiler feedback as a meaningful abstraction of the proof search space, our method reallocates test-time compute toward repairable errors, enabling efficient proof search without long contextual histories or massive roll-outs.

Central to our framework is the view of refinement as a Markovian decision process, where each refinement step depends only on the current failed proof state and verifier feedback. This formulation enables context-efficient iterative refinement and naturally supports adaptive search strategies. In particular, our value-guided refinement policy concentrates exploration in promising regions of the search tree, improving the efficiency of proof discovery under fixed computational budgets.

Extensive experiments demonstrate that the proposed framework consistently amplifies the reasoning capabilities of existing Lean provers across multiple benchmarks and model scales. At the same time, our analysis suggests that refinement effectiveness strongly depends on the structure of the underlying failure mode. While localized and recurring errors are often repairable, deeper semantic failures remain challenging and continue to limit refinement performance.

Although a performance gap still remains relative to large-scale theorem proving systems, our results demonstrate that compiler-guided refinement yields strong gains within fixed-budget single-prover settings, and is complementary to those systematic approaches. Beyond scaling refinement to stronger provers, several promising directions remain open. These include developing more effective search strategies for balancing exploration and refinement, designing improved context management mechanisms for long-horizon reasoning, and learning adaptive refinement policies that dynamically allocate computation across search trajectories.

More broadly, while our current instantiation focuses on symbolic theorem proving, where verification is exact and error messages are structured, it remains unclear how directly the approach transfers to settings with weaker or noisier verification signals. We view formal theorem proving as a particularly suitable testbed for studying verifier-guided refinement, and leave systematic investigation of broader applicability to future work. We believe this work encourages further investigation into principled, feedback-aware search and learning methods for scalable reasoning.

## Acknowledgements

We sincerely thank the reviewers for their thoughtful evaluation and constructive feedback. Their suggestions regarding compiler message injection ablations, training dataset analysis, computation cost analysis, and the Claude ablation study substantially improved the completeness and clarity of this work.

We thank Tianle Hu from Tsinghua university for his assistance with the preparation of the Goedel training and evaluation datasets. We also thank him for valuable early discussions, feedback, and suggestions throughout the project.

This work was supported by the National Natural Science Foundation of China Major Program under Grant 92570203 and the Beijing Natural Science Foundation under Grant Z250001.

## Impact Statement

This work aims to improve the efficiency and scalability of AI systems for formal reasoning. By leveraging compiler feedback to guide theorem proving, our method reduces the computational cost required for automated mathematical verification. More broadly, the proposed framework may contribute to the development of more reliable AI-assisted tools for mathematics, programming, and formal verification. While our experiments focus on theorem proving, the broader applicability of verifier-guided refinement to other domains remains an open direction for future research.

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

# A. Analysis of Compiler Messages

This section provides the distribution of compiler error messages produced by Kimina-Prover-Distill-8B and Goedel-Prover-V2-32B on MiniF2F-test, ProofNet, MathOlympiadBench and PutnamBench. Error messages are grouped after normalization, where variable names, tactic states, and other context-dependent information are masked to enable meaningful aggregation.

*Table 9.* Top 50% error messages across benchmarks. Messages are sorted by ratio within each benchmark.

| KIMINA-PROVER-DISTILL-8B | | GOEDEL-PROVER-V2-32B | |
|---|---|---|---|
| MESSAGE | RATIO (%) | MESSAGE | RATIO (%) |
| **MINIF2F-TEST** | | | |
| LINARITH FAILED TO FIND A CONTRADICTION | 22.60 | UNSOLVED GOALS | 18.87 |
| UNKNOWN IDENTIFIER 'ID' | 16.31 | FAILED TO SYNTHESIZE | 11.48 |
| UNSOLVED GOALS | 13.58 | LINARITH FAILED TO FIND A CONTRADICTION | 9.40 |
| | | UNKNOWN IDENTIFIER 'ID' | 7.84 |
| | | TACTIC 'ID' FAILED, THE LEFT-HAND SIDE | 7.70 |
| **PUTNAM BENCH** | | | |
| TACTIC 'ID' FAILED, NESTED ERROR | 16.78 | TACTIC 'ID' FAILED, NESTED ERROR | 29.77 |
| OMEGA COULD NOT PROVE THE GOAL | 13.50 | UNSOLVED GOALS | 14.84 |
| UNSOLVED GOALS | 11.31 | UNKNOWN IDENTIFIER 'ID' | 8.23 |
| LINARITH FAILED TO FIND A CONTRADICTION | 8.10 | | |
| UNKNOWN IDENTIFIER 'ID' | 5.59 | | |
| **PROOFNET** | | | |
| UNKNOWN IDENTIFIER 'ID' | 15.37 | FUNCTION EXPECTED AT 'ID' | 21.75 |
| UNSOLVED GOALS | 11.34 | FAILED TO SYNTHESIZE | 13.30 |
| FAILED TO SYNTHESIZE | 9.53 | UNKNOWN IDENTIFIER 'ID' | 12.39 |
| UNKNOWN CONSTANT 'ID' | 8.38 | UNKNOWN CONSTANT 'ID' | 9.07 |
| NO GOALS TO BE SOLVED | 6.56 | | |
| **MATHOLYMPIADBENCH** | | | |
| TACTIC 'ID' FAILED, NESTED ERROR | 30.88 | UNSOLVED GOALS | 21.14 |
| OMEGA COULD NOT PROVE THE GOAL | 12.70 | FAILED TO SYNTHESIZE | 16.21 |
| LINARITH FAILED TO FIND A CONTRADICTION | 11.70 | FUNCTION EXPECTED AT 'ID' | 10.13 |
| | | LINARITH FAILED TO FIND A CONTRADICTION | 10.13 |

As clearly demonstrated in Table 9, a few error types dominate most of the failed proof attempts. This concentration pattern is consistent across benchmarks of varying sizes (MiniF2F-test: 244 problems, ProofNet: 371, MoBench: 360, PutnamBench: 660) and across different provers.

This confirms our view of treating the Lean compiler as an effective dimension compressor for the search space of proofs. While the space of syntactically distinct Lean programs is combinatorially large, the compiler maps these programs into a comparatively low-cardinality set of diagnostic categories. As a result, many syntactically different failed proof attempts are collapsed into the same compiler error type, inducing a coarse but stable partition of the program space.

Crucially, this partition is problem-agnostic: the same dominant error types recur across benchmarks and provers, indicating that the compression arises from structural properties of the Lean kernel and tactic language. This many-to-one mapping from programs to compiler messages provides a natural abstraction layer on which refinement policies can operate, enabling generalization across problems by conditioning on error types rather than raw program syntax.

# B. Technical Details of the Proposed Solution

This section provides implementation and prompt-level details necessary for reproducibility.

## B.1. Compiler Message Injection

To guide the provers with structured feedback from the Lean 4 compiler, we insert compiler messages $\Phi(c)$ directly into the source program $c$ according to the reported line numbers in the compiler message. Each compiler message is wrapped by

special boundary tokens `<error>` and `</error>`, which are added to the tokenizer vocabulary of the base provers. No other modification to tokenization is performed. Bodies of compiler messages are treated as plain text and tokenized using the original tokenizer.

Below we show a fragment of Lean 4 program that fails to compile due to an unsuccessful rewrite tactic. The compiler reports a localized error message associated with the corresponding line of Lean program. After inserting the compiler messages, the program is transformed as follows:

```
have h_main :
  (new_set1_mean * (set1_size + 1) + set2_mean * set2_size)
  / (set1_size + 1 + set2_size) = 18.5 := by
  rw [h1, h2, h3, h4, h5]
<error>
tactic 'rewrite' failed, did not find instance of the pattern in the
    target expression
</error>
  norm_num
```

This representation preserves the original program structure while making compiler feedback explicit and locally aligned with the source code. An alternative approach is to append all compiler messages at the end of the program as auxiliary text. However, this obscures the precise location of failures and weakens the association between error messages and the relevant program context.

*Table 10.* Comparison of appending versus inserting compiler messages. *Kimina x2* samples two independent responses from Kimina-Prover-Preview-7B. *<Method>Epoch1/2* denotes the model after 1 or 2 epochs of supervised fine-tuning.

| METHOD | KIMINA X2 | APPEND EPOCH1 | APPEND EPOCH2 | INSERT EPOCH1 | INSERT EPOCH2 |
|---|---|---|---|---|---|
| #SOLVED | 277 | 272 | 246 | 279 | 284 |

We evaluate this design choice through a controlled comparison between two supervision formats: (1) appending compiler messages to the end of the program and (2) inserting them at the corresponding program locations. Both models are fine-tuned from Kimina-Prover-Preview-7B using identical supervised training procedures. Evaluation is conducted on MiniF2F (valid + test, 477 problems) under a fixed inference budget consisting of one direct generation attempt followed by one refinement attempt.

As shown in Table 10, localized insertion consistently outperforms appending. This suggests that spatially aligning compiler diagnostics with the corresponding proof context improves the model's ability to associate verifier feedback with relevant program regions, leading to more effective refinement behavior.

## B.2. Prompt Construction

### B.2.1. DATA SYNTHESIS PROMPT

We use a fixed prompt template to guide Claude 3.7 Sonnet to generate refinement reasoning that explains how to correct compilation errors. The prompt provides a formal problem statement, an incorrect Lean 4 solution augmented with compiler feedback, and a correct reference solution. The model is instructed to produce a reasoning trace that bridges the incorrect and correct solutions, without explicitly revealing the latter. While the refinement target is not constrained to be a minimal edit, conditioning on compiler feedback induces shared corrective structure across problems. An abbreviated version of the prompt is shown below:

```
You are provided with a problem in Lean 4.

I will give you two solutions. The first solution is incorrect, and the
      second solution is correct.
```

```
Your task is to generate a reasoning trace that explains how to fix the
    errors in the first solution using the compiler feedback.

Here is the problem:
{LEAN_PROBLEM}

Here is the incorrect solution. Compiler messages are embedded in the
    code:
{INCORRECT_PROGRAM_WITH_ERRORS}

Here is the correct solution:
{CORRECT_PROGRAM}

Do NOT mention the correct solution in your output. Start from the
    incorrect solution and reason step by step until the errors are
    resolved.

Put your answer inside a <think> </think> block.
```

The generated reasoning pattern is extracted using regular expressions that match the `<think>` and `</think>` delimiters. This synthesized "thought" component is further assembled with the correct Lean program, jointly forming a training data point for supervised fine-tuning. To avoid distributional mismatch and potential training degradation, we strictly follow the original response formats of Kimina-Prover-Distill-8B and Goedel-Prover-V2-32B respectively.

The response template for Kimina-Prover-Distill-8B is shown below:

```
<think>
{SYNTHESIZED_THOUGHT}
</think>

```lean4
{CORRECT_PROGRAM}
```
```

The response template for Goedel-Prover-V2-32B is shown below:

```
### Detailed Proof and Analysis
{SYNTHESIZED_THOUGHT}

### Complete Lean 4 Proof

```lean4
{CORRECT_PROGRAM}
```
```

The input prompt construction is described in the following section.

### B.2.2. PROMPTS FOR DIRECT PROOF GENERATION AND REFINEMENT

For Kimina-Expert, the root node $s_0 = p$ is translated into natural language using the base prover's original prompt template, which induces direct proof generation:

```
Think about and solve the following problem step by step in Lean 4.
'''lean4
{FORMAL_STATEMENT}
'''
```

An intermediate refinement state $s = (c, \Phi(c), p)$ is mapped to the following refinement prompt, which instructs the model to repair the given Lean program using compiler feedback:

```
Think about and fix the following Lean 4 code.
'''lean4
{LEAN_PROGRAM_AND_COMPILER_FEEDBACK}
'''
```

Similarly, for Goedel-Expert, direct proof generation uses the official prompt template:

```
Complete the following Lean 4 code:
'''lean4
{FORMAL_STATEMENT}
'''

Before producing the Lean 4 code to formally prove the given theorem,
    provide a detailed proof plan outlining the main proof steps and
    strategies. The plan should highlight key ideas, intermediate lemmas
    , and proof structures that will guide the construction of the final
     formal proof.
```

For refinement, each intermediate state $s_i$ is translated into the following prompt, which explicitly asks the model to analyze compiler errors and revise the proof accordingly:

```
Fix the following Lean 4 code:
'''lean4
{LEAN_PROGRAM_AND_COMPILER_FEEDBACK}
'''

Before producing the Lean 4 code to formally prove the given theorem,
    provide a detailed proof plan analyzing compiler errors and proof
    steps. The plan should highlight key ideas, intermediate lemmas, and
     proof structures that will guide the construction of the final
    formal proof.
```

### B.3. Training Dataset Analysis

To better understand the nature of the refinement supervision, we manually inspected 200 randomly sampled training examples from each dataset. We categorize each refinement trajectory according to whether the corrected proof substantially builds upon the previous failed attempt or instead restarts from a largely unrelated proof structure.

As shown in Table 11, the vast majority of refinement targets preserve significant portions of the preceding proof attempt. This pattern is consistent across both supervised fine-tuning and expert-iteration datasets for Kimina and Goedel. The results suggest that refinement learning is typically driven by localized repair and modification of existing proof structures rather than complete regeneration from scratch.

*Table 11.* Inpection of the training datasets.

| DATASET | KIMINA SFT | GOEDEL SFT | KIMINA EXPERT ITERATION | GOEDEL EXPERT ITERATION |
|---|---|---|---|---|
| BUILD ON PRIOR | 198 | 190 | 186 | 195 |
| START FROM SCRATCH | 2 | 10 | 14 | 5 |

*Table 12.* Top-20 errors in the training set of Kimina-Prover-Distill-8B.

| MESSAGE | RATIO (%) |
|---|---|
| UNKNOWN IDENTIFIER 'ID' | 40.38 |
| LINARITH FAILED TO FIND A CONTRADICTION | 14.54 |
| UNSOLVED GOALS | 13.43 |
| OMEGA COULD NOT PROVE THE GOAL: | 6.56 |
| TACTIC 'ID' FAILED, DID NOT FIND INSTANCE OF THE PATTERN IN THE TARGET EXPRESSION | 3.84 |
| TYPE MISMATCH | 2.74 |
| TAUTO FAILED TO SOLVE SOME GOALS. | 2.24 |
| SIMP MADE NO PROGRESS | 2.01 |
| NO GOALS TO BE SOLVED | 1.77 |
| TYPE MISMATCH, TERM | 1.19 |
| TACTIC 'ID' FAILED, FAILED TO UNIFY | 1.12 |
| FAILED TO SYNTHESIZE | 0.90 |
| TACTIC 'ID' FAILED, NESTED ERROR: | 0.82 |
| UNKNOWN CONSTANT 'ID' | 0.81 |
| APPLICATION TYPE MISMATCH | 0.67 |
| TACTIC 'ID' FAILED, EQUALITY OR IFF PROOF EXPECTED | 0.59 |
| INVALID PROJECTION, STRUCTURE EXPECTED | 0.58 |
| TACTIC 'ID' EVALUATED THAT THE PROPOSITION | 0.48 |
| TACTIC 'ID' FAILED, INSUFFICIENT NUMBER OF BINDERS | 0.42 |
| FUNCTION EXPECTED AT | 0.40 |
| TOTAL | 95.52 |

This observation supports the central hypothesis underlying compiler-guided refinement: verifier feedback frequently identifies repairable local failure modes for which useful proof structure already exists in the failed attempt. Rather than discarding the previous proof entirely, the refinement process often reuses intermediate lemmas, tactic sequences, or overall proof strategies while modifying the regions associated with verifier failures.

### B.4. Training Error Distribution

Table 12 and Table 13 report the top-20 Lean compiler error types produced by Kimina-Prover-Distill-8B and Goedel-Prover-V2-32B during the data synthesis stage, respectively. These errors arise from unsuccessful proofs generated by the base provers and constitute the primary supervision signal for learning-to-refine. Importantly, the base provers do not natively interpret compiler diagnostics encoded using our error representation, and thus exposure to these error types during training is necessary for acquiring error-conditioned refinement behavior.

To assess generalization, we compare the error types observed during training with those encountered on held-out benchmarks (Table 9). While training and test problems are disjoint, many compiler error categories recur across benchmarks, reflecting structural properties of the Lean proof system and tactic language rather than problem-specific artifacts. This recurrence enables refinement policies to transfer across tasks by operating on shared failure modes, without introducing intentional data leakage.

However, the frequency of an error alone does not determine its repairability. The analysis in Section 5.5 further reveals that refinement is most effective for error types that are both recurrent across problems and admit localized corrective actions. In contrast, errors corresponding to global proof failures or poorly localized semantic gaps remain difficult to repair despite substantial training exposure. Together, these findings support the view that compiler feedback induces a coarse but meaningful partition of failure modes, within which refinement learning selectively generalizes.

*Table 13.* Top-20 errors in the training set of Goedel-Prover-V2-32B.

| GOEDEL-PROVER-V2-32B | RATIO (%) |
|---|---|
| LINARITH FAILED TO FIND A CONTRADICTION | 25.41 |
| UNSOLVED GOALS | 16.41 |
| UNKNOWN IDENTIFIER 'ID' | 8.51 |
| OMEGA COULD NOT PROVE THE GOAL: | 7.78 |
| TACTIC 'ID' FAILED, DID NOT FIND INSTANCE OF THE PATTERN IN THE TARGET EXPRESSION | 6.89 |
| TYPE MISMATCH | 4.23 |
| SIMP MADE NO PROGRESS | 3.59 |
| TACTIC 'ID' FAILED, NESTED ERROR: | 3.08 |
| APPLICATION TYPE MISMATCH | 2.69 |
| FAILED TO SYNTHESIZE | 2.30 |
| NO GOALS TO BE SOLVED | 1.84 |
| UNKNOWN CONSTANT 'ID' | 1.79 |
| TACTIC 'ID' FAILED, MADE NO PROGRESS | 1.40 |
| TYPE MISMATCH, TERM | 1.38 |
| UNEXPECTED TOKEN 'ID'; EXPECTED COMMAND | 1.37 |
| 'ID' HAS ALREADY BEEN DECLARED | 1.31 |
| TACTIC 'ID' FAILED, THE LEFT-HAND SIDE | 0.97 |
| MAXIMUM RECURSION DEPTH HAS BEEN REACHED | 0.87 |
| TACTIC 'ID' FAILED, FAILED TO UNIFY | 0.76 |
| INVALID FIELD 'ID', THE ENVIRONMENT DOES NOT CONTAIN 'ID' | 0.67 |
| TOTAL | 93.26 |

## B.5. Value Function Architecture

Our value functions are based on the resulted Kimina-Expert and Goedel-Expert model respectively. The value function shares the same backbone as the base language model used for proof generation. Given the hidden representation of the final token, the value function uses a value head to produce a scalar score that represents the predicted refinement potential of the corresponding node. Under this design, the value function naturally inputs the same prompt as the prover.

During training, we do not freeze the backbone parameters. This design choice allows the model to adapt its internal representations to better capture long-range dependencies and structural cues that are predictive of successful refinement, which are not necessarily aligned with next-token prediction.

## C. Experiments

### C.1. Distributional Equivalence Test

In a preprocessing step, we canonicalize each Lean program by removing line breaks, redundant whitespace, and in-line annotations, ensuring that edit distances reflect structural edits rather than superficial formatting differences. Let $X = \{x_1, \ldots, x_m\}$ and $Y = \{y_1, \ldots, y_n\}$ denote two sets of canonicalized programs. We define $d(\cdot, \cdot)$ as the length-normalized Levenshtein edit distance between two strings, given by

$$d(x, y) = \frac{\text{Lev}(x, y)}{\max(|x|, |y|)},$$

where $\text{Lev}(\cdot, \cdot)$ denotes the standard Levenshtein distance and $|\cdot|$ the string length. This normalization controls program length and yields a bounded distance in $[0, 1]$. While edit distance is a purely syntactic metric, Lean proofs exhibit tight coupling between syntactic structure and proof state evolution. Thus, even localized semantic changes typically induce nontrivial edits in the canonicalized code.

To compare the distributions induced by two program generation processes, we employ the energy distance statistic. The empirical energy statistic between $X$ and $Y$ is defined as

$$E(X, Y) = \frac{2}{mn} \sum_{i=1}^{m} \sum_{j=1}^{n} d(x_i, y_j) - \frac{1}{m^2} \sum_{i=1}^{m} \sum_{j=1}^{m} d(x_i, x_j) - \frac{1}{n^2} \sum_{i=1}^{n} \sum_{j=1}^{n} d(y_i, y_j),$$

*Table 14.* Top-10 errors produced by Kimina-Expert under random search strategy. Frequencies in training set are reported in Appendix B.4.

| ERROR TYPE | OCCUR FREQ (%) | #PROBLEM | FIX PROB (%) | TRAIN FREQ (%) |
|---|---|---|---|---|
| OMEGA COULD NOT PROVE THE GOAL: | 40.30 | 31 | 0.04 | 6.56 |
| LINARITH FAILED TO FIND A CONTRADICTION | 19.54 | 48 | **0.12** | **14.54** |
| UNSOLVED GOALS | 17.43 | 49 | **0.03** | **13.43** |
| TYPE MISMATCH, TERM | 2.85 | 20 | 0.00 | 1.19 |
| TACTIC 'ID' FAILED, DID NOT FIND INSTANCE OF | 2.39 | 39 | 0.24 | 3.84 |
| THE PATTERN IN THE TARGET EXPRESSION | | | | |
| UNKNOWN CONSTANT 'ID' | 2.32 | 28 | 0.00 | 0.81 |
| UNKNOWN IDENTIFIER 'ID' | 1.87 | 25 | 0.30 | 40.38 |
| TACTIC 'ID' FAILED, FAILED TO UNIFY | 1.81 | 25 | 0.00 | 1.12 |
| FAILED TO SYNTHESIZE | 1.67 | 12 | **2.02** | **0.90** |
| NO GOALS TO BE SOLVED | 1.61 | 25 | **0.35** | **1.77** |

We assess distributional equivalence using a permutation-based hypothesis test. The null hypothesis is

$$H_0 : X \overset{d}{=} Y$$

i.e., the two samples are exchangeable and drawn from the same underlying distribution. Given observed samples $A$ and $B$, we compute the observed statistic $E_0 = E(A, B)$. We then construct an empirical null distribution by repeatedly permuting the pooled sample $C = A \cup B$. For each of $N_{\text{perm}}$ permutations, $C$ is randomly partitioned into subsets $A_k$ and $B_k$ with $|A_k| = |A|$ and $|B_k| = |B|$, and the corresponding statistic $E_{\text{perm}} = E(A_k, B_k)$ is evaluated. The $p$-value is estimated as the fraction of permutation statistics that are at least as large as the observed statistic:

$$\hat{p} = \frac{1 + \#\{E_{\text{perm}} \geq E_0\}}{1 + N_{\text{perm}}}.$$

which yields an unbiased estimate under finite permutations. This test is used in Section 5.4 to compare programs produced by breadth-first search and iterative refinement on MiniF2F-test-hard, and consistently rejects the null hypothesis, indicating that compiler-guided refinement induces a distribution over programs that is distinct from that of direct generation.

### C.2. Error-Type Analysis

Table 14 reports error-type statistics for Kimina-Expert under random tree search, analogous to those presented for Goedel-Expert in Table 7. For each error type, we report its occurrence frequency, the number of distinct problems in which it appears, the empirical probability of correction by refinement, and its relative frequency in the refinement training data.

Overall, Kimina-Expert exhibits trends consistent with those observed for Goedel-Expert. Error types that admit localized corrective actions (e.g., `failed to synthesize`) achieve higher correction probabilities, while errors corresponding to deep semantic reasoning or poorly localized failures remain difficult to repair despite substantial training exposure. All error types with nonzero correction probability appear across many distinct problems and are present in the training distribution, indicating that refinement generalizes over recurring failure modes rather than memorizing problem-specific solutions.

### C.3. Discussion on Claude

One may question whether the improvements in our framework primarily originate from the additional capability of a frontier model such as Claude-3.7-Sonnet. Our setting differs from standard distillation: both failed and correct proofs are generated by the base provers (Kimina/Goedel), while Claude is used only to produce natural-language refinement explanations linking them. In particular, Claude does not directly provide executable repair solutions beyond the capability of the base prover.

To isolate the effect of Claude-generated refinement traces, we conduct a controlled comparison on MiniF2F (valid + test, 477 problems) using Kimina-Prover-Preview-7B under a fixed inference budget of 2. The three settings differ only in the generation strategy for the second attempt: (1) independent resampling, (2) Claude-assisted refinement without refinement fine-tuning, and (3) self-refinement using the supervised fine-tuned (SFT) refinement model. Results are shown in Table 15.

Claude-assisted refinement yields only modest improvements over independent sampling, whereas the SFT refinement model

*Table 15.* Ablation study on refinement strategies with Claude.

| METHOD | KIMINA X2 | KIMINA + CLAUDE | KIMINA SFT + REFINE |
|---|---|---|---|
| #SOLVED | 277 | 280 | 284 |

achieves substantially larger gains. This suggests that the primary benefit does not arise from directly leveraging a stronger external model at inference time, but rather from learning reusable self-repair behaviors from refinement trajectories.

More importantly, the improvement from SFT alone remains relatively limited compared with the gains obtained through iterative refinement search. To study this effect, we compare cold-start supervised fine-tuning against expert iteration on Goedel-V2 under the same random refinement strategy.

*Table 16.* Comparison between supervised fine-tuning and expert iteration.

| MODEL | STRATEGY | MINIF2F-TEST | PROOFNET | MOBENCH | PUTNAMBENCH |
|---|---|---|---|---|---|
| GOEDEL-V2 | DIRECT | 84.43% | 15.63% | 14.72% | 32 SOLVED |
| GOEDEL-V2 | RANDOM | 84.02% | 26.42% | 25.56% | 46 SOLVED |
| GOEDEL SFT | RANDOM | 84.84% | 26.42% | 26.11% | 47 SOLVED |
| GOEDEL EXPERT | RANDOM | 84.43% | 23.72% | 30.28% | 63 SOLVED |

As shown in Table 16, cold-start SFT provides only marginal improvements over the base refinement strategy, whereas expert iteration contributes the majority of the final performance gains, particularly on more difficult benchmarks such as PutnamBench and MOBench. This suggests that the dominant factor is not the presence of Claude-generated reasoning traces themselves, but the iterative refinement process induced by on-policy exploration and self-correction.

One possible interpretation is that cold-start refinement traces primarily teach generic local repair behaviors, while expert iteration exposes the model to progressively harder and more diverse failure modes encountered during actual refinement search. As a result, the refinement policy becomes increasingly adapted to the distribution of intermediate proof states encountered at inference time.

## D. Case Studies

### D.1. Prompts and Responses during the Refinement Process

For each refinement state $s = (c, \Phi(c), p)$, the prompt is constructed using previously explained template. Below we show an example problem sourced from Goedel-LM/Goedel-Pset-v1. During the refinement process, our Kimina-Expert encounters a failed proof attempt:

```lean4
Think about and fix the following Lean 4 code.
```lean4
import Mathlib
import Aesop

set_option maxHeartbeats 0

open BigOperators Real Nat Topology Rat

/-- Show that for any natural number $n$, 12 divides $4^{n+1} + 20$.-/
theorem induction_12dvd4expnp1p20 (n : ℕ) : 12 | 4 ^ (n + 1) + 20 := by
  have h1 : 4 ^ (n + 1) + 20 ≡ 0 [MOD 12] := by
    have h2 : 4 ^ (n + 1) ≡ 4 [MOD 12] := by
      have h3 : 4 ^ (n + 1) % 12 = 4 := by
        have h4 : n + 1 ≥ 1 := by omega
        have h5 : 4 ^ (n + 1) % 12 = 4 := by
          induction (n + 1) with
          | zero => simp
<error>
unsolved goals
case zero
```

```
n : ℕ
h4 : n + 1 ≥ 1
⊢ False
</error>
            | succ m ih =>
              rw [Nat.pow_succ, Nat.mul_mod, ih]
              simp
<error>
no goals to be solved
</error>
        exact h5
      exact h3
    have h6 : 20 % 12 = 8 := by norm_num
    have h7 : (4 ^ (n + 1) + 20) % 12 = 0 := by
      simp [Nat.add_mod, h2, h6]
<error>
unsolved goals
n : ℕ
h2 : 4 ^ (n + 1) ≡ 4 [MOD 12]
h6 : 20 % 12 = 8
⊢ (4 ^ (n + 1) % 12 % 12 + 8 % 12) % 12 = 0
</error>
    exact h7
  exact Nat.dvd_of_mod_eq_zero h1
```

The response of Kimina-Expert is:

```
<think>
Looking at the error messages, I can see that there are two main issues
    :

1. In the 'zero' case of the induction for 'h5', we have 'n : ℕ' and '
   h4 : n + 1 ≥ 1', but we're trying to prove 'False'. This doesn't
   make sense - we shouldn't be trying to prove 'False' here.

2. In the 'succ m ih' case, after applying 'rw [Nat.pow_succ, Nat.
   mul_mod, ih]', we get 'no goals to be solved', but then we're still
   trying to apply 'simp'. This suggests that the rewrite might have
   already solved the goal, or that 'simp' is being applied in a
   context where there are no goals.

Let me think about what's happening mathematically. We want to prove
    that '4 ^ (n + 1) % 12 = 4' for all 'n + 1 ≥ 1'.

- Base case: When 'n + 1 = 1' (so 'n = 0'), we need to show '4 ^ 1 % 12
    = 4', which is true since '4 % 12 = 4'.
- Inductive step: Assume '4 ^ (m) % 12 = 4' (induction hypothesis 'ih')
    . We need to show '4 ^ (m + 1) % 12 = 4'.

Using 'Nat.pow_succ', we have '4 ^ (m + 1) = 4 ^ m * 4'. Then using '
    Nat.mul_mod', we get '(4 ^ m * 4) % 12 = (4 ^ m % 12 * 4 % 12) %
    12'. By the induction hypothesis, '4 ^ m % 12 = 4', so we get '(4 *
    4) % 12 = 16 % 12 = 4'. This is what we want.

The issue seems to be that the 'simp' tactic is either solving the goal
     too early or not applying correctly. Let me reconsider the approach
     .

Instead of using induction on 'n + 1', maybe I should use a different
```

```
    approach. Looking at the pattern, '4 ^ (n + 1) % 12 = 4' seems to
        hold for all 'n + 1 ≥ 1'. Let me try a more direct approach using
        properties of modular arithmetic.

    Actually, wait. Let me reconsider the overall proof structure. Instead
        of proving '4 ^ (n + 1) ≡ 4 [MOD 12]' separately, maybe I can
        directly prove '(4 ^ (n + 1) + 20) % 12 = 0' using induction on 'n'.

    For the base case 'n = 0', we need to show '(4 ^ 1 + 20) % 12 = 0',
        which is '(4 + 20) % 12 = 24 % 12 = 0'.

    For the inductive step, assuming '(4 ^ (k + 1) + 20) % 12 = 0', we need
        to show '(4 ^ (k + 2) + 20) % 12 = 0'.

    We have '4 ^ (k + 2) = 4 * 4 ^ (k + 1)'. So '4 ^ (k + 2) + 20 = 4 * 4 ^
        (k + 1) + 20'. Using the induction hypothesis, we know '4 ^ (k + 1)
        + 20 ≡ 0 [MOD 12]', which means '4 ^ (k + 1) ≡ -20 ≡ -8 ≡ 4 [MOD
        12]'.

    Then '4 * 4 ^ (k + 1) ≡ 4 * 4 = 16 ≡ 4 [MOD 12]'. So '4 * 4 ^ (k + 1) +
        20 ≡ 4 + 20 = 24 ≡ 0 [MOD 12]'.

    This approach seems cleaner and avoids the issues with the original
        proof. Let me try restructuring the proof this way.
</think>
```lean4
import Mathlib
import Aesop

set_option maxHeartbeats 0

open BigOperators Real Nat Topology Rat

/-- Show that for any natural number $n$, 12 divides $4^{n+1} + 20$.-/
theorem induction_12dvd4expnp1p20 (n : ℕ) : 12 ∣ 4 ^ (n + 1) + 20 := by
  have h1 : (4 ^ (n + 1) + 20) % 12 = 0 := by
    have h2 : ∀ (n : ℕ), (4 ^ (n + 1) + 20) % 12 = 0 := by
      intro n
      induction n with
      | zero =>
        norm_num
      | succ n ih =>
        have h3 : (4 ^ ((n + 1) + 1) + 20) % 12 = 0 := by
          have h4 : 4 ^ ((n + 1) + 1) = 4 * (4 ^ (n + 1)) := by
            ring_nf
          rw [h4]
          simp [Nat.add_mod, Nat.mul_mod] at ih ⊢
          all_goals
            omega
        exact h3
    specialize h2 n
    simpa using h2
  exact Nat.dvd_of_mod_eq_zero h1
```
```

The generated Lean 4 program successfully passes the compilation check, indicating that the refinement step yields a valid proof. This example illustrates that Kimina-Expert has acquired the ability to condition its generation on compiler feedback, a capability that is absent from the base prover. Examining the failed proof attempt in detail, the compiler emits three diagnostic messages corresponding to two error types: unsolved goals and no goals to be solved. According to the error-type statistics in Table 14, these error categories have relatively low empirical correction probabilities (0.03% and 0.35%, respectively). As discussed in Section 5.5, such error types typically reflect global or poorly localized

proof failures rather than isolated tactic-level mistakes.

This characterization is consistent with the present example, where the failure arises from an ill-posed global induction structure. In such cases, no immediate local repair is available, and successful refinement requires proposing a different proof strategy conditioned on the failed attempt. Nevertheless, this example also highlights an important nuance: while global errors are difficult to correct on average, their correctability can depend strongly on the underlying problem instance. For relatively easy problems such as this one, the effective probability of successful correction may be higher than suggested by aggregate statistics computed over challenging benchmarks such as MiniF2F-test-hard.

### D.2. Visualizations of Test-Time Search Behaviors

#### D.2.1. DIRECT PROOF GENERATION AND ITERATIVE REFINEMENT

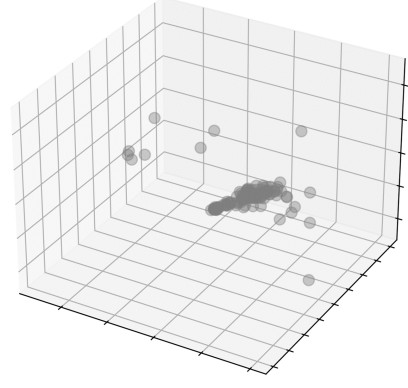 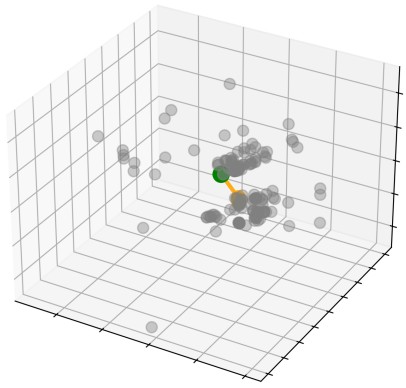

*(a)* Direct solution generation produces a dense cluster.      *(b)* Successful iterative refinement trajectory.

*Figure 3.* Visualization of programs generated by Kimina-Expert on problem Putnam-1965-b5. Each gray circle represents a failed Lean proof in program space. Green circle corresponds to a correct solution. The successful search trajectory is highlighted in orange.

As analyzed quantitatively in Section 5.4, for most problems, the direct generation distribution $D_{\text{direct}}(\cdot \mid p)$ and the refinement-conditioned distributions $\{D_{\text{refine}}(\cdot \mid c_i, \Phi(c_i), p)\}_{i=0}^{n}$ are statistically distinct in Lean program space. To provide an intuitive geometric perspective on this separation, we further visualize the sampled Lean programs using multidimensional scaling (MDS) (Mead, 2018).

The MDS embeddings are computed from the full pairwise edit-distance matrix over programs sampled from both distributions. As shown in Figure 3a, programs generated via direct proof sampling form a compact cluster in the embedded space. As the sampling budget increases, the diameter of this cluster grows only gradually, indicating limited expansion of the explored region of program space.

In contrast, Figure 3b illustrates that iterative refinement induces a sequence of refinement states that progressively moves away from the initial cluster. Visually, we can clearly observe that the traced trajectory exits the support of the direct-generation distribution and explores new regions of the space. While MDS provides only a low-dimensional approximation of the underlying geometry, this visualization is consistent with our chain-of-distributions hypothesis: refinement operates by transitioning between distinct distributions rather than by densifying a single global distribution.

#### D.2.2. RANDOM AND VALUE-GUIDED SEARCH

Figure 4 illustrates the behavior of random tree search and value-guided search on the problem `Putnam-1971-b1`. The search tree is generated by Goedel-Expert, and the search process terminates once a verified proof is found. Under a random expansion strategy, the prover discovers a correct solution after exploring 45 nodes in the search tree. In contrast, value-guided search finds a proof after only 5 node expansions.

As shown in Figure 4b, the learned value function assigns the highest value to the root node relative to its immediate descendants. As a consequence, the value-guided search policy allocates a larger fraction of the sampling budget to expanding the root, effectively prioritizing direct proof generation over early refinement steps in this instance. While

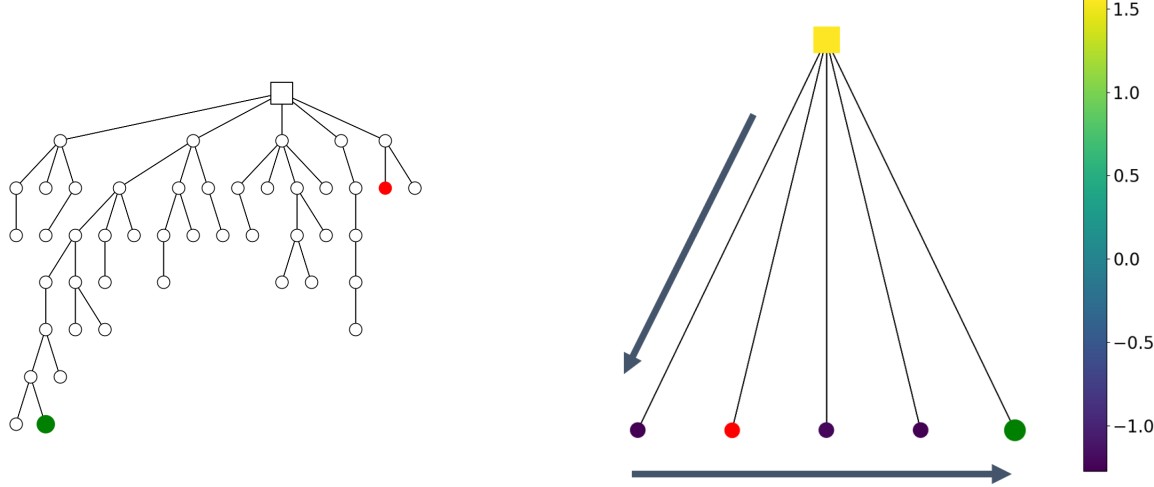

*(a)* Random tree search strategy.           *(b)* Value-guided tree search strategy.

*Figure 4.* Visualization of random tree search strategy and value-guided tree search on problem Putnam-1971-b1 under Goedel-Expert. Figure 4b visualizes the values for each tree node using colormap. Red nodes correspond to broken nodes (where LLM's proof generation process failed). Green nodes represent correct solutions.

both strategies eventually succeed, value-guided search achieves substantially greater sample efficiency by concentrating exploration on states that the value model predicts to be closer to a complete proof.

This example highlights that value guidance does not uniformly favor refinement: rather, it adaptively balances direct generation and refinement based on the predicted utility of intermediate states. In this case, the model correctly identifies that the root state already admits a high-probability direct solution, leading to a more budget-efficient search strategy. More generally, this behavior suggests that value-guided search can recover classical direct generation as a special case, while retaining the ability to invoke refinement when intermediate states are predicted to be more promising.

