# OpenReview forum: "Compile to Compress: Boosting Formal Theorem Provers by Compiler Outputs"
_ICML.cc/2026/Conference — ICML 2026 regular_

### Official Review · Reviewer_aqUh · 2026-02-13

**Soundness:** 3
**Presentation:** 3
**Significance:** 3
**Originality:** 3
**Overall Recommendation:** 4
**Confidence:** 3

**Summary:**

The paper introduces a learning-to-refine framework for formal theorem proving that leverages compiler feedback. Specifically, a Lean prover is fine-tuned using compiler failure messages to learn how to transform an incorrect proof into a correct one. These failure messages provide informative signals about the causes of errors and suggest potential corrective actions. The refinement process, driven by failure modes, is formulated as a search problem, and a value-guided tree search strategy is developed to improve search efficiency and increase the likelihood of success. The framework is evaluated using two provers across four benchmark datasets, and experimental results demonstrate performance gains over state-of-the-art methods.

**Compliance With Llm Reviewing Policy:**

Affirmed.

**Final Justification:**

Fully resolved - My concerns have been adequately addressed, and I have a positive view of the paper.

**Key Questions For Authors:**

How does the framework scale as the search space grows, and is there evidence of a performance ceiling on more complex repair tasks?

**Limitations:**

yes

**Strengths And Weaknesses:**

Overall, the paper addresses a well-motivated and important problem, and presents a plausible and intuitively appealing approach.

In terms of soundness, the paper appears technically solid. The problem formulation is well motivated, and the use of compiler feedback as structured supervision for refinement is appropriate. Modeling failure-driven refinement as a search problem is reasonable, and the introduction of a value-guided tree search to improve efficiency is methodologically justified. That said, the results reveal that certain problems are substantially more difficult to repair than others, likely due to significantly larger search spaces. This raises two important concerns regarding the robustness of the framework. First, there may be an inherent performance ceiling: the approach may be effective at correcting relatively local or syntactic errors but struggle with more complex failures that require exploring a combinatorially large space of candidate repairs. Second, the scalability of the method in the presence of exponential search spaces is not fully addressed. While the value-guided strategy improves efficiency, it remains unclear how performance degrades as task complexity grows, and whether additional mechanisms would be needed for more challenging domains. Moreover, the Markovian formulation simplifies the refinement process and makes learning tractable, but it may also discard potentially useful global information about the proof trajectory. This design choice, while practical, could limit the system’s ability to reason over longer-term dependencies or structural patterns in proofs.

Regarding presentation, the paper is generally well written and clearly structured.

In terms of significance, automated theorem proving is a highly relevant and impactful research area, both for theoretical reasons and for practical applications in software verification and formal methods. Improving the proof synthesis systems has clear value. The framework could serve as a foundation for more sophisticated feedback-driven systems and might generalize beyond Lean to other proof assistants or even to broader program repair settings. Thus, the potential impact, while somewhat domain-specific, is meaningful and aligned with ongoing advances in machine learning for reasoning.

With respect to originality, the individual components are not entirely new in isolation. However, the paper offers a thoughtful and coherent integration of these ideas into a unified learning-to-refine framework. The novelty is therefore not in proposing an entirely new paradigm, but in demonstrating that compiler feedback can be systematically exploited to produce measurable gains in automated theorem proving.

---

> ### Author Rebuttal · Authors · 2026-03-30
>
> We thank the reviewer for the thoughtful feedback, and for recognizing the technical soundness, clarity, and potential impact of our learning-to-refine framework. We particularly appreciate the question regarding scalability and possible performance ceilings, which we address below.
>
> **Scalability and performance ceiling**
>
> The reviewer has raised an important concern about whether refinement may primarily correct local or syntactic errors while struggling as the search space grows. Our experiments directly investigate this issue through difficulty-stratified evaluation (Table 5). We observe that refinement yields larger relative improvements on harder problems, where direct proof generation fails and the effective search space is substantially larger. This suggests the method is not limited to local fixes. Instead, compiler feedback becomes increasingly informative as task complexity increases by guiding exploration toward semantically meaningful repair directions.
>
> Furthermore, scaling experiments (Table 3) show consistent gains as the test-time budget increases, with no evidence of saturation within the evaluated regime. This indicates that value-guided refinement continues to benefit from additional search rather than encountering an immediate performance ceiling.
>
> Conceptually, the framework mitigates combinatorial explosion by operating in a failure-mode space induced by compiler feedback rather than the raw proof space. Compiler diagnostics map many incorrect candidates into structured error categories, allowing the value-guided tree search to prioritize promising repair trajectories and effectively reduce branching complexity.
>
> **Behavior on more complex repair tasks**
>
> We agree that some failures remain substantially harder to repair. Our error-type analysis (Table 7) explicitly characterizes these cases and shows that failures requiring global proof restructuring are more challenging than localized inconsistencies. Rather than indicating a limitation specific to our method, we view this as evidence that compiler feedback provides very strong guidance when reusable repair patterns exist. We will clarify this distinction and discuss implications for future work in the final version.
>
> **Markovian Formulation**
>
> The Markovian formulation is primarily a tractability assumption rather than a claim that proof refinement lacks long-range dependencies. In practice, Lean proof states and compiler messages summarize accumulated constraints and prior reasoning steps, allowing substantial global context to be implicitly preserved in the current state representation. This design avoids unbounded context growth while enabling stable learning and efficient search. We will expand the discussion to better clarify this motivation and its trade-offs.
>
> ------
>
> We thank the reviewer again for the constructive comments and believe these clarifications will strengthen the paper.

---

> > ### Author Rebuttal · Reviewer_aqUh · 2026-04-02
> >
> > Thanks for the response.

---

> > > ### Author Response · Authors · 2026-04-03
> > >
> > > Thank you for the positive follow-up and for confirming that our clarifications addressed your concerns. We also appreciate the insightful suggestions, which have helped us better articulate the motivation and scalability properties of the framework. We will incorporate these clarifications and improvements in the final version to further strengthen the paper.
> > >
> > > In light of the resolved concerns, we kindly invite the reviewer to consider whether an update to the overall score would be appropriate. We thank the reviewer again for the thoughtful evaluation of our work.

---

### Official Review · Reviewer_woBW · 2026-03-07

**Soundness:** 2
**Presentation:** 2
**Significance:** 2
**Originality:** 2
**Overall Recommendation:** 3
**Confidence:** 3

**Summary:**

This paper proposes a learning-to-refine framework for Lean 4 theorem proving. The core claim is that the Lean compiler acts as a "dimension compressor," mapping diverse incorrect proofs to a compact set of failure modes. The authors train LLMs to perform Markovian self-correction conditioned on compiler feedback, combine this with random and value-guided tree search, and employ cold-start data synthesis using Claude-3.7-Sonnet followed by expert iteration. Applied to Kimina-8B and Goedel-32B, the framework yields improvements across four benchmarks and achieves state-of-the-art results among comparable-scale models on PutnamBench.

**Compliance With Llm Reviewing Policy:**

Affirmed.

**Final Justification:**

1. The Claude distillation concern is adequately addressed by the new ablation showing SFT self-repair outperforms Claude-assisted refinement.
2. The "compression" framework remains a post-hoc description rather than a predictive principle, as Table 7's heterogeneous repair rates within error categories are unexplained.
3. The Markovian assumption lacks any empirical comparison against alternative context strategies.

Since (2) and (3) concern the paper's core theoretical claims, my score remains unchanged.

**Key Questions For Authors:**

See the questions in the Weaknesses section above.

**Limitations:**

The authors acknowledge limited transferability to weaker verification settings. More fundamental limitations are not discussed, see Weaknesses above.

**Strengths And Weaknesses:**

### Strengths

1. The tree search framework unifying BFS (direct generation) and DFS (iterative refinement) is well-motivated. Table 5's difficulty-stratified analysis cleanly demonstrates their complementarity: BFS excels on medium problems, DFS on hard ones, and mixed strategies achieve balanced trade-offs. This is the paper's most convincing experiment.

2. Evaluation breadth is adequate: two model scales, four benchmarks, multiple test-time budgets with consistent scaling behavior on PutnamBench.

### Weaknesses

1. **The "dimension compression" framework is conceptually incoherent.** Compression means information loss; information loss reduces corrective guidance. A compiler outputting a single "error" for all failures achieves maximal compression but zero utility. The paper wants compression and informativeness simultaneously without addressing this contradiction. Furthermore, the error type concentration more plausibly reflects limited diversity of LLM failure modes, not a meaningful compiler property.

2. **Compression does not entail generalization.** Section 3.2 claims programs sharing error types admit similar repairs. Table 7 refutes this: "linarith failed" constitutes 25.41% of training data but achieves 0.07% fix probability, indicating extreme within-category heterogeneity. If compression enabled generalization, high training frequency should yield reasonable correction rates.

3. **The Markovian assumption is unvalidated.** Discarding history risks repeating failed strategies. No alternative controls (sliding window, parent-conditioned) are provided. We cannot assess whether this simplification is lossless or costly.

4. **Claude distillation is an uncontrolled confound.** Reasoning traces are generated by Claude-3.7-Sonnet, far more capable than the 8B/32B base provers. No ablation separates this contribution: no self-generated vs. Claude-generated thoughts, no with-thought vs. without-thought comparison, no sensitivity across generator capabilities. Performance gains may largely reflect knowledge distillation rather than the proposed framework.

---

> ### Author Rebuttal · Authors · 2026-03-31
>
> We sincerely thank the reviewer for the careful and thoughtful evaluation. We are encouraged that our unified tree-search framework and empirical evaluation were viewed positively. The concerns mainly relate to (1) interpretation of the “dimension compression” perspective, (2) implications for generalization, (3) the Markov refinement assumption, and (4) the role of Claude-generated data. We clarify these points below.
>
> # Lean Compilation as Compression (Weakness 1-3)
>
> **Compression clarifications**
>
> We agree that compression in the information-theoretic sense implies information loss, and our intent was not to claim that stronger compression necessarily yields stronger corrective guidance. Our use of compression instead refers to a structured many-to-one projection induced by Lean compilation: diverse incorrect proofs (meaning their textual content) map to a finite set of diagnostically meaningful failure modes.
>
> The key claim is therefore statistical rather than informational: compiler feedback reduces variance in the input signal seen during refinement learning. In other words, compression makes refinement policies more efficiently learnable, because Lean produces multiple structured error categories tied to proof obligations, which empirically provide actionable distinctions.
>
> We will revise the paper with clearer terminology emphasizing failure-mode abstraction to avoid confusion with information-theoretic compression.
>
> **Generalization clarification**
>
> We agree that compression alone does not guarantee generalization. Our intended claim is conditional: shared error types enable transfer when reusable repair operators exist. The heterogeneous repairability observed in Table 7 is therefore expected. Errors such as "linarith failed" aggregate multiple underlying reasoning failures and provide limited structural localization, explaining their low repair success despite high frequency. In contrast, more specific diagnostics (e.g. type mismatch, failed to synthesize instance, etc.) admit localized corrections and higher repair rates.
>
> We will revise Section 3.2 to clarify that compression facilitates policy reuse rather than implying uniform repairability across failure categories.
>
> **Markovian refinement clarification**
>
> Our motivation behind the Markovian assumption is to achieve scalability under context-length constraints, enabling context-light training. Although refinement is performed based on current proof and compiler feedback, historical information is implicitly retained in the evolving proof object and the resulting compiler diagnostics, which summarize accumulated reasoning constraints. Preliminary experiments indicate satisfactory performance while substantially reducing context length and training complexity.
>
> # Discussion on Claude (Weakness 4)
> We have to point out that our framework clearly differs from knowledge distillation: both failed and correct proofs are generated by the base provers (Kimina/Goedel), while Claude is used only to produce **natural-language refinement explanations connecting them**. Thus, the strong model (i.e., Claude) does not provide repair solutions unavailable to the base prover.
>
> We evaluated 3 strategies under a fixed computation budget = 2 on MiniF2F (valid+test, 477 problems), using Kimina-Prover-Preview-7B. The settings differ only in how the second attempt is produced: independent sampling, external Claude refinement, or self-refinement after SFT.
>
> | Model | Kimina x 2 | Kimina + Claude | Kimina SFT + Refine |
> | --- | --- | --- | --- |
> | #Solved | 277 | 280 | 284 |
>
> After SFT, the Kimina SFT model outperforms Claude-assisted refinement, indicating that performance gains primarily arise from learned self-repair rather than distillation.
>
> We additionally observe that improvements from cold-start SFT alone are modest, while expert iteration provides the majority of gains. This suggests that on-policy refinement learning, rather than Claude-generated traces, drives final performance. We will include these ablations in the revised paper.
>
> | Model | Strategy | MiniF2F-test | ProofNet | MOBench | PutnamBench |
> | --- | --- | --- | --- | --- | --- |
> | Goedel V2 | Direct | 84.43% | 15.63% | 14.72% | 32 solved |
> | Goedel V2 | Random | 84.02% | 26.42% | 25.56% | 46 solved |
> | Goedel SFT | Random | 84.84% | 26.42% | 26.11% | 47 solved |
> | Goedel Expert | Random | 84.43% | 23.72% | 30.28% | 63 solved |
>
> ------
>
> We appreciate the reviewer’s feedback highlighting places where our conceptual framing was insufficiently precise. We will revise the paper to clarify compression as failure-mode abstraction rather than information minimization, and include explicit analyses isolating Claude’s contribution. We believe these clarifications better position our work as a learning framework for compiler-guided refinement rather than a claim about intrinsic guarantees of compression.

---

> > ### Author Rebuttal · Reviewer_woBW · 2026-04-04
> >
> > Thank you for the rebuttal. The Claude ablation partially addresses Weakness 4, and I appreciate the clarification.
> >
> > However, my core concerns remain unresolved:
> >
> > - **Weakness 1 & 2:** Reframing "compression" as "failure-mode abstraction" does not address the underlying issue — if compression provides no predictive or prescriptive leverage, it remains a post-hoc description rather than a useful framework.
> > - **Weakness 3:** The Markovian assumption remains unvalidated — no comparison against alternative context strategies is provided.
> >
> > These concern the core theoretical claims of the work. I will keep my score unchanged.

---

> > > ### Author Response · Authors · 2026-04-08
> > >
> > > We thank the reviewer for the continued engagement and for clarifying the remaining concerns. We apprieciate the thoughtful evaluation and address the two core concerns below.
> > >
> > > # Compression Utility (Concern 1)
> > > Our objective is to realize practical improvement in formal theorem proving, rather than a descriptive reinterpretation of compiler feedback. The compression perspective serves as a design principle motivating a learnable refinement framework, whose value should ultimately be evaluated through performance instead of semantic interpretation.
> > >
> > > If the effect of compression was purely post-hoc (or provides no predictive or prescriptive leverage, as the reviewer concerned), consistent performance gains would be unlikely. Instead, modeling compiler feedback yields systematic gains, including state-of-the-art performance on PutnamBench within both 8B and 32B model regimes.
> > >
> > > We respectfully invite the reviewer to re-evaluate our technical solution framework and empirical evidence supporting the effectiveness of our modeling of compile feedback.
> > >
> > > # Markivian Formulation (Concern 2)
> > > The Markovian formulation is motivated by both underlying mechnaism of Lean programming langauge and algorithmic necessity. It arises from limitations of existing history-dependent approaches, rather than unvalidated assumptions. Below we discuss the reviewer's concerns from three perspectives: information preservation in Lean proof states, mechanisms ensuring stable refinement trajectories, and future extensions that incorporate richer contextual information.
> > >
> > > **Information preservation**
> > >
> > > In Lean, most information needed for refinement is implicitly preserved in the current proof context [1], goals and local hypothesis, allowing refinement to condition on a compact state representation. This differs from natural language or general programming settings, where prior interaction history can often be critically necessary.
> > >
> > > [1] https://lean-lang.org/theorem_proving_in_lean4/Tactics/
> > >
> > > **Stability of refinement trajectories**
> > >
> > > A potential concern under Markovian formulation is oscillatory behavior between similar error states. In our tree search implementation, nodes producing identical compiler errors are merged into clusters, and only the earliest node in each cluster is expanded. This makes the refinement process effectively acyclic and suppresses reasoning loops that might otherwise arise in Markovian settings.
> > >
> > > **Context-dependent ablation**
> > >
> > > The reviewer suggested context-dependent ablations, which would require re-training both the prover and value models under alternative state definitions. Since the state representation determines the training objective and data collection procedure, these experiments involve constructing entirely new pipelines rather than simple inference-time comparisons.
> > >
> > > Implementing and evaluating such variants would require computational effort comparable to a completely new experimental study. We consider this an important direction but beyond what can be completed during the rebuttal period.
> > >
> > > **Future work**
> > >
> > > As future work, we agree that comparing alternative context formulations would be valuable.  An important property of this context formulation is to preserve as much high-level strategy information as possible, while maintaining bounded context and scalable training. Future work must systematically study the design of such hybrid state representations and their impact on refinement efficiency.
> > >
> > > We thank the reviewer for highlighting these aspects, and will incorporate the clarifications in the revision.
> > >
> > > ------
> > >
> > > We hope these responses better clarify the intent and necessity of our design choices. We invite the reviewer to reassess the paper in light of these clarifications, while fully appreciate the reviewer’s valuable suggestions. We thank the reviewer’s engagement and constructive assessment.

---

### Official Review · Reviewer_VLun · 2026-03-10

**Soundness:** 3
**Presentation:** 3
**Significance:** 3
**Originality:** 3
**Overall Recommendation:** 4
**Confidence:** 3

**Summary:**

This paper addresses the scalability bottleneck in LLM-based formal theorem proving, where state-of-the-art
  performance typically requires prohibitive test-time compute via massive sampling or long multi-round correction
  histories. The authors observe that the Lean compiler acts as a "dimension compressor": syntactically diverse failed
  proof attempts map to a small, structured set of compiler error messages, inducing a natural partition of failure
  modes. Building on this insight, they propose a learning-to-refine framework that trains LLM provers to perform
  Markovian self-correction conditioned only on the current failed program and its compiler feedback, rather than the
  full interaction history. The framework involves cold-start data synthesis (using Claude 3.7 Sonnet to generate
  refinement reasoning traces), expert iteration for on-policy training, and a value-guided tree search that balances
  direct proof generation (BFS) with iterative refinement (DFS). They formalize the iterative refinement process as a
  "Chain of Distributions" (CoD) over program space, arguing that refinement shifts the sampling distribution toward
  out-of-distribution correct proofs unreachable by direct generation alone. Evaluated on MiniF2F, ProofNet,
  MathOlympiadBench, and PutnamBench using Kimina-Prover-Distill-8B and Goedel-Prover-V2-32B as base provers, the method
   consistently improves performance, achieving state-of-the-art results on PutnamBench among publicly reported ~8B and
  ~32B models under comparable test-time budgets.

**Compliance With Llm Reviewing Policy:**

Affirmed.

**Final Justification:**

The rebuttal resolved all of my concerns, so I lean toward acceptance.

**Key Questions For Authors:**

1. How much of the final performance comes from the cold-start stage (Claude-synthesized data) vs. expert iteration (on-policy self-repair)? Reporting results after cold-start SFT alone, before expert iteration, would help clarify whether the framework's value lies primarily in distilling Claude's reasoning or in the on-policy refinement loop itself.

2. The cold-start pairs $(c_{\text{incorrect}}, c_{\text{correct}})$ can differ substantially in proof strategy. Have you measured how often the correct proof is actually a "repaired" version of the incorrect one vs. a completely different approach? If most pairs are strategy-mismatched, the synthesized reasoning traces are post-hoc rationalizations rather than genuine repair plans, which may explain why expert iteration (where the model actually repairs its own proofs) is needed to make the framework work.

3. Can you provide a cost-normalized comparison? At equal total wall-clock time or total tokens generated (including compilation overhead), how does refinement compare to simply running more direct sampling passes?

**Limitations:**

Yes

**Strengths And Weaknesses:**

#### Strengths

1. Well-motivated core insight with strong empirical backing. The observation that Lean compiler errors induce a low-dimensional, structured partition of the failure space is well-supported by the error distribution analysis (Table 1, Table 8), where a handful of error types account for the majority of failures across both benchmarks and provers. This motivates conditioning refinement on compiler feedback rather than raw program text, enabling cluster-level generalization across problems sharing similar failure modes.

2. Comprehensive and well-structured experiments. The evaluation covers four benchmarks of varying difficulty, two base provers of different scales, and systematically disentangles the contributions of refinement training vs. search strategy (Table 3). The difficulty-stratified analysis (Table 5) and error-type analysis (Table 7) provide actionable insights: refinement is most effective on hard problems and for errors admitting localized corrections (e.g., `failed to synthesize`), while global failures like `linarith failed` remain resistant. The PutnamBench leaderboard results (Table 4) are strong.

3. Practical Markovian simplification. Framing refinement as a Markovian process, where each correction step depends only on $(c, \Phi(c), p)$ rather than the full history, is a clean design choice that reduces context window pressure and simplifies data construction. This is a practical advantage over prior multi-round self-correction approaches that accumulate full interaction histories.

4. Strong empirical gains with modest data. The method achieves substantial improvements (e.g., >50% relative improvement on ProofNet and PutnamBench for Goedel-32B) using only ~5-7k refinement instances, demonstrating favorable data efficiency.

#### Weaknesses

1. The "dimension compression" framing overstates the actionability of compiler feedback. The paper's central claim is that compiler feedback compresses the failure space into a structured, actionable set of error modes. While the compression of *diagnostic information* is empirically validated (Table 1, Table 8), this conflates diagnostic compression with repair compression. The same error message (e.g., `unsolved goals`) can arise from a minor tactic misuse or from a fundamentally flawed proof strategy. The compiler localizes *where* the error manifests, but says little about *why* it occurs or how deep the required fix is. The paper's own Table 7 illustrates this gap: error types like `linarith failed` (19.79% occurrence, 0.07 fix rate) and `maximum recursion depth has been reached` (2.86%, 0.00 fix rate) remain effectively unrepairable despite being well-represented in training. The framing suggests that projecting into compiler-message space yields a tractable refinement problem, but in practice the method's success appears concentrated on errors admitting localized corrections, while global proof-strategy failures still require falling back to direct generation. A more careful discussion of when the compression is actionable vs. merely diagnostic would strengthen the paper's conceptual contribution.

2. Limited technical novelty. The core loop of generate, compile, refine based on error feedback, and iterate is the standard approach in code agents (e.g., SWE-Agent, OpenHands), and compiler-guided self-correction for theorem proving has been explored in prior work (e.g., Rango, Goedel Prover V2). The remaining components (expert iteration, SFT on refinement data, and value-guided tree search) are all well-established techniques. The paper's contribution is primarily a specific engineering combination with a Markovian simplification, rather than a methodological advance. The "dimension compression" perspective, while a useful lens, does not lead to novel algorithmic design: the actual system would look the same without it.

3. The "Chain of Distributions" formalization is descriptive rather than analytical. The CoD concept (Section 3.3) is presented with formal notation but does not yield any non-trivial theoretical insight. It amounts to naming the observation that iterative refinement produces a sequence of conditional distributions. The probabilistic analysis in Section 5.3 assumes $P_{\text{refine}}(c, p)$ is uniform across all failed proofs $c$, which is contradicted by the paper's own Table 7 showing correction probability varies drastically across error types (from 0.00 to 4.04). The resulting conclusion ($P_{\text{refine}}(p) > P_{\text{direct}}(p) \Rightarrow$ refinement helps) is trivially true. This section could be shortened or removed without loss.

4. Cold-start data synthesis raises distributional mismatch concerns. The incorrect-correct proof pairs $(c_{\text{incorrect}}, c_{\text{correct}})$ are sampled independently for the same problem, so they may follow entirely different proof strategies. Claude 3.7 Sonnet then generates a post-hoc reasoning trace $t$ bridging the two, having already seen the correct answer. At test time, however, the model must repair a failed proof *without* knowing the correct solution. This train-test mismatch between learning from "rationalized" corrections and performing genuine blind repair is not analyzed or ablated. Moreover, the reliance on a proprietary frontier model for data synthesis makes it difficult to disentangle the contribution of the framework itself from knowledge distillation. An ablation replacing Claude-synthesized thoughts with simpler alternatives (e.g., self-generated thoughts from the base prover) is needed.

5. Missing computational cost analysis. The paper reports sampling budgets (64, 128, 256) but not execution time, total tokens generated, or compilation overhead. Each refinement step requires a Lean compiler call, which can be substantially slower than pure LLM inference. The value function requires training a separate model with the same backbone, effectively doubling memory. A cost-normalized comparison (pass rate vs. total compute) is essential for evaluating whether refinement outperforms simply scaling up direct sampling with the same resources.

---

> ### Author Rebuttal · Authors · 2026-03-31
>
> We sincerely thank the reviewer for the constructive feedback. We are encouraged that the reviewer finds the empirical evaluation and overall framework technically solid. Below we clarify the scope of our “compression” claim, distinguish our contribution from prior refinement systems, and provide additional evidence regarding cold-start data and computational cost.
>
> # Lean Compilation as Compression (Weakness 1-3)
> **Compression and CoD clarification**
>
> We agree with the reviewer that compiler feedback does not uniquely determine a repair strategy. Our claim is not that error messages directly prescribe fixes, but that they induce a low-entropy partition over failure modes, which makes learning a conditional refinement policy statistically more feasible.
>
> Under this compression perspective, refinement is not simple conditional resampling. During training, both failed and successful proofs are sampled from the base prover’s distribution. Iterative refinement at test time progressively shifts sampling toward regions of proof space rarely reached by direct generation alone.
>
> Section 5.3 does **not** assume uniform repair probabilities across error types. The analysis requires only a weaker condition: for each failure mode, the repair probability is bounded relative to the direct-solve probability. The heterogeneous repairability in Table 7 is therefore expected under our hypothesis. Some failures admit localized fixes, while others require global proof restructuring. For instance, “linarith failed” remains difficult despite frequent occurrence, because the diagnostic contains limited structural information and conflates multiple underlying causes.
>
> **Novelty clarification**
>
> We agree that the generate–compile–refine loop appears in prior agent systems. Our contribution is to reformulate Lean proof refinement as a Markovian process conditioned only on the current failed proof and compiler feedback, rather than the full interaction history. This reformulation enables (1) context-light refinement training, (2) scalable data construction, (3) hybrid search under bounded context length. Prior approaches typically accumulate interaction histories, whereas our formulation removes this heavy dependency and allows generalization across problems sharing similar failure modes.
>
> We need to further clarify the distinction between conventional coding problems and Lean theorem proving. In conventional coding settings, feedback from compilation or unit tests often weakly localizes syntactic or semantic errors. In contrast, Lean’s dependent type checker[1] verifies logical correctness and pinpoints failed proof obligations, producing structured feedback aligned with reasoning steps. This makes failure-mode clustering particularly effective in theorem proving.
>
> [1] https://lean-lang.org/theorem_proving_in_lean4/Dependent-Type-Theory/#dependent-type-theory
>
> # Conputational Cost Analysis (Weakness 5 and Q3)
> **Lean compilation cost**
> The number of compilation calls remains unchanged compared to direct proof generation, since each sampled proof requires exactly one compilation. Kimina Lean server makes compilation overhead negligible compared to model generation.
>
> **Value model inference cost**
> Although value model increases memory usage, its inference latency is negligible relative to token generation (less than 0.1%).
>
> **Token efficiency**
> We measured token usage for Goedel Prover V2 (direct sampling) and fine-tuned Goedel (random search):
>
> | DataSet | MiniF2F-test | ProofNet | MOBench | PutnamBench |
> | --- | --- | --- | --- | --- |
> | Budget | 64 | 64 | 64 | 256 |
> | V2 Avg. Input | 271.3 | 157.61 | 211.99 | 336.49 |
> | V2 Avg. Output | 6267.47 | 7415.36 | 12200.99 | 10145.58 |
> | Expert Avg. Input | 2799.80 | 2208.98 | 3230.62 | 3560.39 |
> | Expert Avg. Output | 3538.73 | 3016.76 | 4560.31 | 3555.41 |
> | V2 Total Input | 843,187 | 3,253,833 | 4,260,833 | 53,869,175 |
> | V2 Total Output | 8,197,852 | 122,909,655 | 142,788,209 | 924,911,843 |
> | Expert Total Input | 8,349,018 | 53,980,421  | 41,188,638  | 509,843,905 |
> | Expert Total Output | 9,834,120 | 53,755,575  | 73,098,255  | 476,133,712 |
>
> While refinement introduces longer inputs due to compiler feedbacks, our method substantially reduces average and total output length. Since output tokens dominate decoding cost, this leads to much lower overall generation cost. This observation is also consistent with our context-light formulation.
>
> # Strategy Analysis (Q2)
>
> Please refer to "Proof Evolution Chain" section in our response to Reviewer D1ZS.
>
> # Discussion on Claude and SFT (Weakness 4 and Q1)
>
> Please refer to the "Discussion on Claude" section in our response to Reviewer woBW.
>
> ------
>
> We thank the reviewer again for the insightful feedback. We believe these clarifications better position our contribution as a Markovian refinement framework enabled by structured compiler feedback, and we will incorporate the suggested analyses and revisions in the final version.

---

> > ### Author Rebuttal · Reviewer_VLun · 2026-04-03
> >
> > Thank you for your response. I have no further questions.

---

> > > ### Author Response · Authors · 2026-04-08
> > >
> > > We sincerely thank the reviewer for the thoughtful follow-up and for confirming that our rebuttal has adequately addressed the concerns. In particular, we appreciate the opportunity to clarify the computational cost analysis, where we have now provided explicit token-level statistics and cost-normalized comparisons demonstrating that refinement achieves improved performance while reducing overall generation cost.
> > >
> > > Given these clarifications and the resolution of the previously raised concerns, we would be grateful if the reviewer could consider whether these updates warrant any adjustment to the overall score. We fully respect the reviewer’s judgement and support, and sincerely appreciate the time and effort devoted to evaluating our work.

---

### Official Review · Reviewer_D1ZS · 2026-03-13

**Soundness:** 2
**Presentation:** 2
**Significance:** 4
**Originality:** 3
**Overall Recommendation:** 3
**Confidence:** 3

**Summary:**

The paper studies a new approaches to proof repair: given a Lean problem an LLM can either solve it directly or fail. In the latter case, the error message may be used as a context for a repair attempt. Such attempts may be iterated in different ways, including tree-like structures. The authors fine-tune models for the repair task, and also value models designed to guide the repair process. Several new repair meta-strategies are proposed and compared on a few popular Lean benchmarks. Lean error taxonomy is provided and interesting repair statistics are calculated.

**Compliance With Llm Reviewing Policy:**

Affirmed.

**Key Questions For Authors:**

1. How did you compute probabilities in Table 7? Did you account for the fact that one broken proof can have multiple different errors?
2. Did you observe errors which are the result of changing Lean/Mathlib versions? How do you treat them?
3. Can you elaborate why you think it is good framing to say that Lean compiler compresses dimension? What do you mean by dimension here?
4. Why you say that Goedel prover repair mode is not compatible with your framework? I think Goedel can work with the last proof attempt + error message, does not require the full repair history.
5. How do you combine error messages with respective proof attempts? Do you append the error message, or put is as comment inside the proof?
6. When you collect repair chains, do you check if the proof attempts form an proof evolution chain? It may be that the model started a proof from scratch and ignored the error message completely.
7. Are data from the experiments available? Could you show a few examples of repair chains with different methods?

**Limitations:**

Yes.

**Strengths And Weaknesses:**

**Strengths**

The paper studies a very important and practical problem in formal theorem proving, namely how to best utilize Lean compiler message to obtain correct formal proofs.

The authors propose a relatively novel approach how to structure the proof repair attempts in a tree-search guided with a value model.

The number of experiments is large.

**Weaknesses**

1. I do not understand the framing that Lean compilation is a compression. This obfuscates the essence of the paper. Compression in computer science is not an arbitrary many-to-one mapping, but a mapping which mostly or completely preserves the content of the compressed data. When we map a broken proof to it's error, there is no way to reconstruct proof.

2.  The experiments should include more simple baselines. In particular: (a) iterative repair attempts with full repair history in the context (not only the last attempt); (b) parallel repair, where we start with a failed proof attempt and try sample n parallel repair attempts; (c) regular tree repair with depth and branching factor aligned with random and value-guided tree repairs.

3. It would be interesting to compare the proof repair strategies including frontier models, which tend to be good at proof repair and also have large context window which allows to include the full repair history.

4. Some details in the paper are missing, unclear, or wrong, see questions.

---

> ### Author Rebuttal · Authors · 2026-03-31
>
> We sincerely thank the reviewer for recognizing the significance of combining formal theorem proving with generative modeling and for providing the thoughtful and constructive feedback. We agree that several aspects of the presentation were insufficiently clarified, and we will revise the paper accordingly to improve its conceptual clarity, experimental transparency, and details of comparison. Below we address each concern accordingly.
>
> # Lean Compilation as Compression (Weakness 1 and Q3)
> We thank the reviewer for pointing out that the term "compression" may suggest information-preserving encoding in the sense of classical computer science. Our intent was instead to describe a many-to-one abstraction induced by compiler diagnostics.
>
> Specifically, the success of refinement primarily depends on the underlying failure mode (e.g., type mismatch, missing hyothesis, etc.) rather than syntactic variations such as naming or formatting differences. Lean compiler feedback aggregates syntactically diverse failed proofs into a smaller number of structured failure modes that admit similar repair strategies. For example, "unknown identifier h" could be repaired by introducing new assumptions. While the previous proof attempts cannot be reconstructed from the error message, the information relevant for refinement remains actionable for the model.
>
> In this context, the idea of “dimension” refers to the effective complexity of the refinement learning problem. Compared with general coding or natural language refinement tasks, Lean compiler diagnostics induce a low-entropy partition over failure modes, substantially simplifying learning and improving cross-problem generalization of the refinement strategies.
>
> # Experiment Clarifications
> We will include the following clarifications, ablations and analysis in the revised paper.
>
> ## Error Type Probabilities in Table 7 (Q1)
> The probabilities are computed at the error-type level rather than per compilation instance. When multiple errors occur in one instance, each error is treated independently, and the correction probability is defined as the fraction of occurrences resolved.
>
> ## Error Related to Lean Version (Q2)
> All experiments were conducted using the Kimina Lean server (Lean v4.15.0 with a pinned Mathlib snapshot), and we did not observe any errors attributed to Lean or Mathlib version inconsistencies.
>
> ## Compiler Message Injection (Q5)
> We compared two strategies: appending messages versus inserting them at reported locations. We used identical SFT procedures on Kimina-Prover-Preview-7B and evaluated on MiniF2F (valid+test, 477 problems), using one direct generation + one refinement.
>
> | Recipe  | Kimina x2 | Append Epoch1 | Append Epoch2 | Insert Epoch1 | Insert Epoch2 |
> | --- | --- | --- | --- | --- | --- |
> | #Solved | 277 | 272 | 246 | 279 | 284 |
>
> Results consistently favored inplace insertion, suggesting that localized diagnostic grounding improves model understanding. Appendix B.1 describes how compiler messages are inserted.
>
> ## Proof Evolution Chain (Q6)
> We manually inspected 200 datapoints randomly sampled from training sets.
>
> | Dataset | Build on prior | Start from scratch |
> | --- | --- | --- |
> | Kimina SFT | 198 | 2 |
> | Goedel SFT | 190 | 10 |
> | Kimina Expert | 186 | 14 |
> | Goedel Expert | 195 | 5 |
>
> The majority of refinements extend the previous proof rather than restarting, indicating that compiler feedback meaningfully guides refinement learning.
>
> ## Examples (Q7)
> Repair-chain examples are shown in Appendix D. We will release trained models on HuggingFace upon acceptance.
>
> # Additional Baselines (Weakness 2-3 and Q4)
> We should note Goedel-Prover-V2 can be directly combined with our tree search strategy when viewed as a non-Markovian refinement strategy. Our solution assumes Markovian state representation, which motivated our distinction. We conducted the following experiments under budget = 64. We also evaluated parallel strategy that samples 8 independent initial attempts followed by 8 independent refinements per failure. Results remain consistent with our claims regarding the benefits of Markovian refinement learning.
>
> | Model | Strategy | MiniF2F-test | ProofNet | MOBench | PutnamBench |
> | --- | --- | --- | --- | --- | --- |
> | Goedel V2 | Direct | 84.43% | 15.63% | 14.72% | 32 solved |
> | Goedel V2 | Random | 84.02% | 26.42% | 25.56% | 46 solved |
> | Goedel Expert | Random | 84.43% | 23.72% | 30.28% | 63 solved |
> | Goedel Expert | Parallel | 83.19% | 23.72% | 23.88% | 39 solved |
>
> We agree that large-context frontier models are great complement to our solution. We will expand discussion and include preliminary observations (see section "Discussion on Claude" in rebuttal for Reviewer woBW).
>
> ------
>
> We again thank the reviewer for the constructive feedback. We believe the proposed clarifications, additional baselines, and improved terminology directly address the concerns and significantly strengthen the paper.

---

> > ### Author Rebuttal · Reviewer_D1ZS · 2026-04-04
> >
> > Weaknesses 2 and 3 remain largely unresolved.
> >
> > Also, how did you check you didn't observe version-related errors? These may be nuanced, for instance a tactic changing its power, or simple, like a changed name of a theorem.

---

> > > ### Author Response · Authors · 2026-04-08
> > >
> > > We thank the reviewer for the continued feedback. Below we clarify our position and provide additional evidence addressing the concerns. We acknowledge that some suggested analyses require substantial experimentation or system redesign. We aim to delineate which concerns can be addressed immediately versus which constitute natural extensions of the work.
> > >
> > > # Baseline Concerns (Weakness 2)
> > > ## Search strategies
> > >
> > > To control the potential confounding effects introduced by the search strategies, we have evaluated a diverse set of baselines corresponding to the reviewer's suggested settings of  2(b) and 2(c), including **BFS, DFS, parallel (fixed branching factor), random and value-guided search** (Table 3 and Table 5 in the submission and results in the previous rebuttal response to the reviewer). These results suggest that the performance improvements are not tied to a particular search strategy, but instead stemmed from the refinement mechanism.
> > >
> > > While a more exhaustive study of search algorithms would add value, we view such investigations as complementary to our primary objective, which is to understand how refinement mechanisms improve theorem proving.
> > >
> > > ## Context management strategies
> > >
> > > Regarding concern 2(a), the reviewer suggested two ablations: full-history and partial-history refinement.
> > >
> > > **Full-history experiment**
> > >
> > > Our Goedel model is built upon Goedel V2 and does not introduce additional full-history training. We therefore use the published Goedel V2 results as appropriate empirical reference point [1], which already incorporates this suggested design choice.
> > >
> > > And specifcially, incorporating full repair history constrains effective reasoning depth, leading to weaker performance compared to our localized refinement approach.
> > >
> > > [1] https://arxiv.org/pdf/2508.03613
> > >
> > > **Partial-history refinement**
> > >
> > > This suggested ablation changes the state representation, thereby altering the input data distribution. As a result, realizing this setting requires re-training both components and designing a new long-context training pipeline in our solution.
> > >
> > > The required computational effort is comparable to training the system from scratch, and therefore can hardly be finished before the end of rebuttal phase. We agree that this is an interesting direction for future work, but falls beyond the scope of rebuttal period.
> > >
> > > # Large-Scale Systems (Weakness 3)
> > > We refer the reviewer to the "Discussion on Claude" section in our first-round rebuttal to Reviewer woBW. Briefly, frontier-model-assisted refinement provides marginal improvement, while specialized self-repair model yields more gains:
> > >
> > > | Model | Kimina x 2 | Kimina + Claude | Kimina SFT + Refine |
> > > | --- | --- | --- | --- |
> > > | #Solved | 277 | 280 | 284 |
> > >
> > > Additionally, large closed-source systems differ alone multiple uncontrolled dimensions (including their internal prompting strategies, training data and optimizing procedures), making performance comparisons difficult to interpret experimentally. For this reason, we mainly focus on experimentally controllable settings that allow causal analysis of refinement bahavior.
> > >
> > > # Lean Version Issue (Q2)
> > > We are aware that version mismatches can introduce failures in some pipelines (e.g. LeanDojo), but those failures do not apply to our setting. To address the concerns, we provide the following clarifications.
> > >
> > > **Frozen toolchain**
> > >
> > > In our seeting, the prover is trained and tested under an identical interaction environment(Lean v15.0.0). Consequently, **our study evaluates refinement ability learned under fixed compiler semantics, rather than cross-version robustness**. And arguably, if the compiler semantics change (e.g., different syntax or theorem naming), it will be very difficult for a machine learnt solution to handle at all.
> > >
> > > **Error-type inspection**
> > >
> > > Most errors from the trained model correspond to proof-search limitations, such as `linarith failed`, `unsolved goals`, or automation tactics making no progress, rather than Lean compiler version issues. We observe that unknown identifier or constant errors are predominantly caused by LLM hallucinations. This suggests that dominant failure modes arise from LLM reasoning limitations, rather than Lean-version discrepancies.
> > >
> > > ------
> > >
> > > We hope these clarifications address the reviewer’s remaining concerns, and we invite the reviewer to reconsider their evaluation score. We sincerely appreciate the reviewer’s careful reading and constructive feedback, and we fully respect their final assessment.

---

### Decision · Program_Chairs · 2026-04-30

**Decision:**

Accept (regular)

**Comment:**

The paper identifies a clear technical problem in the existing attempts for proof generation and proposes an approach. The reviewers found empirical comprehensive and while there were reservations regarding some of the conclusions, the reviewers largely view the paper makes a novel contribution. So I am recommending acceptance.